## [Peer Review File · The EMBO Journal]

The lysosomal LAMTOR-Rag complex functions as a checkpoint for antiviral interferon production

Zeming Feng, Lulu Wang, Shujun Chen, Sihan Cao, Miao Lei, Xiuzhen Yang, Kaixiong Ma, Shi YU, Huina Hu, Kaixuan Zheng, Xin Xu, Qi Zheng, Shaobo Wang, Wenxiang Hu, and Chun-Yan Lim

Corresponding author: Chun-Yan Lim (lin_junyan@gzlab.ac.cn)

Review Timeline:

Submission Date:	23rd Sep 25
Editorial Decision:	30th Oct 25
Revision Received:	26th Nov 25
Editorial Decision:	18th Dec 25
Revision Received:	29th Dec 25
Accepted:	5th Jan 26

Editor: Ieva Gailite

Transaction Report:

Please note that the manuscript was previously reviewed at another journal. As EMBO Press has a transfer agreement with that journal, revision was invited based on the reports from that previous external review.

We sincerely thank the reviewers for their valuable insights and constructive suggestions, which have significantly strengthened our manuscript. Below, we provide detailed point-by-point responses to all comments.

Reviewers' comments:

Reviewer #1 (Remarks to the Author):

In this study, Feng et al. describe a non-canonical role for the lysosomal LAMTOR-Rag signaling hub in the regulation of the interferon response upon infection or activation of innate immune pathways. The authors show that the LAMTOR/Rag-induced expression of IFNbeta involves changes in the expression levels of specific IRFs and is independent of innate immune recognition and of mTORC1 signaling. Instead, this response is strongly influenced by the activation state of the Rag dimer (determined by the Rag nucleotide loading), and requires the activation of p38 MAPK on lysosomes, where it is recruited in a Rag-dependent manner.

The main findings described in this manuscript are novel and interesting, and most claims are supported by good quality data. The manuscript is very well written and the data presentation is excellent. The involvement of the LAMTOR-Rag-p38 machinery in IFN induction is robustly demonstrated in multiple cell lines and by using conditional Rag KO mice.

In contrast, a more detailed analysis of the mechanisms that control the lysosomal recruitment and activation of p38 is missing from the current version of the manuscript and would substantially strengthen this study. Moreover, additional controls would be necessary to properly characterize the genetically modified cell lines in terms of the effects of the perturbations in LAMTOR/Rag/Raptor on mTORC1 signaling, and to formally exclude a potential role for mTORC1-TFEB/TFE3 signaling in the IFN response. Specific comments and suggestions to improve the manuscript further are listed below.

Dear Reviewer #1,

We sincerely appreciate your positive evaluation of our manuscript and your constructive suggestions for improving our study. In the revised manuscript, we have addressed your key concerns by:

1. Strengthening the epistatic relationship between p38 MAPK and LAMTOR-Rag and elucidating the mechanisms of p38 lysosomal recruitment

To delineate the epistasis between p38 MAPK and LAMTOR-Rag, we generated *Fln* KO and *Depdc5* KO cells, which exhibit opposing Rag GTPase nucleotide states. Consistent with the loss of LAMTOR-Rag, FLCN depletion abolished p38 MAPK phosphorylation and the IFN response. Conversely, DEPDC5-deficient cells displayed enhanced p38 phosphorylation and a stronger IFN response (**new Figure 5a-c**). Crucially, ectopic expression of RagA-GTP or RagC-GDP in *Mapk14/Mapk11* DKO cells did not restore the IFN program (**Figure 5d-g**). These findings strongly suggest that p38 MAPK functions downstream of LAMTOR-Rag GTPases to regulate IFN induction.

Figure 5

Fig. 5 | Rag nucleotide-loading state controls p38 MAPK activation necessary for IFN induction.

a, Immunoblot analysis of p38 MAPK phosphorylation in *Ragc* KO, *Fln* KO, and *Depdc5* KO cells compared with parental cells.

b, *Ifnb1* mRNA expression in parental, *Fln* KO, and *Depdc5* KO cells after PRR stimulation. Data are mean \pm SD, n = 3 biological replicates per group; two-way ANOVA followed by Tukey's test, adjusted P value as indicated.

c, IFN- β secretion from parental, *Fln* KO, and *Depdc5* KO cells after PRR stimulation, as measured by ELISA. Data are mean \pm SD, n = 3 biological replicates per group; two-way ANOVA followed by Tukey's test, adjusted P value as indicated.

d, Ectopic expression of FLAG-tagged RagA-GTP (Q66L) or RagC-GDP (S74N) in *Mapk14/11* DKO cells.

e, *Ifnb1* mRNA expression in *Mapk14/11* DKO cells expressing FLAG-tagged RagA-GTP or RagC-GDP after PRR stimulation. Data are mean \pm SD, n = 3 biological replicates per group; two-way ANOVA followed by Tukey's test, adjusted P value as indicated.

f, IFN- β secretion from *Mapk14/11* DKO cells expressing FLAG-tagged RagA-GTP or RagC-GDP after PRR stimulation. Data are mean \pm SD, n = 3 biological replicates per group; two-way ANOVA followed by Tukey's test, adjusted P value as indicated.

g, Schematic showing epistasis across the GATOR1 and FLCN-FNIP1/2 complex, RagA/C GTPases, and p38 MAPK.

Since the distinct Rag GTPase nucleotide states specifically affected p38 MAPK phosphorylation but not its downstream nuclear targets ATF-2 and c-Jun (**Figure 4j in the revised manuscript**), we examined p38 MAPK subcellular localization following PRR stimulation. Nuclear-cytoplasmic fractionation assays revealed predominant cytoplasmic localization of p38 MAPK, regardless of PRR stimulation or Lamtor1 status (**new Figure 6a**). Using polymer-based magnetic bead isolation, we demonstrated the presence of p38 MAPK in phagolysosomal fractions (**Figure 6b,c**).

Figure 6

Fig. 6 | p38 MAPK is required to regulate *Ifnb1* mRNA stability.

a, Nuclear-cytoplasmic fractionation assays to determine the p38 MAPK localization in response to PRR stimulation.

b, Schematic workflow for the isolation of native phagolysosomes.

c, Immunoblot analysis of phagolysosomes isolated from parental and *Lamtor1* KO cells. WCL, whole cell lysate.

d, Immunoblot analysis of phagolysosomes isolated from parental, *Flcn* KO, and *Depdc5* KO cells. The asterisk (*) indicates a non-specific band detected by the FLCN antibody.

e, Interaction between mBaoJin-MAPK14 and FLAG-tagged mouse FLCN or the indicated variants, as assessed by FLAG immunoprecipitation in HEK293T cells.

f, Representative micrographs showing co-localization of mBaoJin-MAPK14 with LysoView-positive lysosomes. Scale bars: 10 μ m.

Building on our findings that FLCN and DEPDC5 oppositely regulate p38 phosphorylation and IFN responses (**Figure 5a–c**), we tested whether Rag GTPase nucleotide states control p38 lysosomal recruitment. *Depdc5* knockout markedly increased phagolysosomal localization of both Raptor and p38, while *Flcn* knockout abolished p38 phagolysosomal association (**Figure 6d**). Co-immunoprecipitation

demonstrated a specific interaction between p38 MAPK and FLCN, requiring both Longin and DENN domains (**Figure 6e**). Supporting these observations, live-cell imaging revealed abolished co-localization of mBaoJin-MAPK14 with LysoView-positive lysosomes in *Fln* KO versus *Depdc5* KO or control cells (**Figure 6f**).

Collectively, these data identify FLCN as a key mediator of p38 MAPK lysosomal recruitment, directly coupling Rag GTPase nucleotide status to spatially regulated p38 activation.

2. Demonstrating the specific role of p38 in IFN induction, which is to regulate *Ifnb1* mRNA transcript stability.

To corroborate the key role of p38 in IFN induction, we depleted *Mapk14* and/or *Mapk11* from cells using CRISPR. Loss of either or both isoforms impaired *Ifnb1* transcription and IFN- β secretion (**Extended Data Figure 10a,b; Figure 5e,f**), establishing p38's involvement in regulating IFN responses. Based on these findings, we have removed the p38 MAPK chemical inhibition data (**previously Figure 8a-d and Extended data Figure 9d-f**) from the revised manuscript.

Extended Data Figure 10

Extended Data Fig. 10 | p38 MAPK activation is critical for PRR-induced IFN induction.

a,b, *Ifnb1* transcription (**a**) and IFN- β release (**b**) from parental, *Mapk14* KO, and *Mapk11* KO cells after PRR stimulation. Data are mean \pm SD, $n = 3$ biological replicates per group; two-way ANOVA followed by Tukey's test, adjusted P value as indicated.

We then assessed whether p38 kinase activity was required for this regulation. Overexpression of a kinase-dead p38 mutant (D168A) significantly suppressed IFN responses, whereas a constitutively active mutant (D176A/F327S) did not further enhance responses beyond wild-type levels (**Extended Data Figure 10f,g**). These data suggest p38 activity is necessary but not limiting.

Extended Data Figure 10

Extended Data Fig. 10 | p38 MAPK activation is critical for PRR-induced IFN induction.

f,g, *lfnb1* transcription (**f**) and IFN- β release (**g**) from parental cells or cells expressing MAPK14 WT, D168A, or D176A/F327S after PRR stimulation. Data are mean \pm SD, $n = 3$ biological replicates per group; two-way ANOVA followed by Tukey's test, adjusted P value as indicated.

Considering the inherent instability of *lfnb1* mRNA, we hypothesized that extranuclear p38 MAPK activity during PRR stimulation might regulate its stability. We employed Roadblock-qPCR to measure the half-life of *lfnb1* mRNA. Remarkably, *lfnb1* mRNA degradation was significantly accelerated in both *Lamtor1* KO and *Mapk14/Mapk11* DKO cells relative to parental cells (**Figure 6g**). In contrast, DEPDC5 deficiency substantially prolonged *lfnb1* mRNA half-life, an effect that was reversed by *Mapk14/Mapk11* deletion (**Figure 6h**). These results demonstrate that p38 MAPK critically regulates *lfnb1* mRNA stability downstream of the LAMTOR-Rag GTPases.

Figure 6

Fig. 6 | p38 MAPK is required to regulate *Ifnb1* mRNA stability.

g, Measurements of *Ifnb1* mRNA decay using Roadblock-qPCR in parental, *Lamtor1* KO, and *Mapk14/11* DKO cells.

h, Measurements of *Ifnb1* mRNA decay using Roadblock-qPCR in parental, *Depdc5* KO, *Mapk14/11* DKO, and *Depdc5/Mapk14/Mapk11* TKO cells.

i, Representative micrographs of RAW macrophages of the indicated genotypes following infection with IAV PR8-NS1-GFP at an MOI of 0.25 for 24 h. Scale bars: 100 μ m.

Consistent with this post-transcriptional role of p38, impaired IFN production in *Lamtor1* KO, *Ficn* KO, and *Mapk14/11* DKO cells increased viral susceptibility, whereas the potent antiviral state in *Depdc5* KO cells was strictly dependent on p38 MAPK activity (Figure 6i).

3. Establishing that the LAMTOR-Rag complex modulates IFN production independently of the mTORC1-TFEB/TFE3 axis.

TFEB and TFE3 are noncanonical mTORC1 phosphorylation substrates regulated by the LAMTOR-Rag axis on lysosomal membranes. Specifically, RagC and its cognate GAP FLCN are essential for mTORC1-mediated phosphorylation of TFEB/TFE3. FLCN promotes the GDP-bound state of RagC, enabling RagC to bind TFEB/TFE3 and facilitate their phosphorylation by mTORC1 [PMID: 36697823, 32612235]. Consequently, disruption of the LAMTOR-Rag axis or loss of FLCN leads to TFEB/TFE3 dephosphorylation and nuclear translocation, enhancing their transcriptional activities.

Consistent with this regulatory mechanism, we validated the phosphorylation dynamics of TFEB following deletion of these components in RAW macrophages (Reviewer Figure 1). In contrast to the loss of *Lamtor1* or FLCN, DEPDC5 deficiency resulted in sustained TFEB phosphorylation, independent of nutritional status.

Reviewer Figure 1

Reviewer Figure 1 | Nutrient-dependent TFEB phosphorylation in RAW macrophages. Immunoblot analysis of TFEB phosphorylation status in parental, *Lamtor1* KO, *Flcn* KO, and *Depdc5* KO cells under amino acid starvation and refeeding conditions.

To determine whether TFEB/TFE3 suppresses IFN transcription downstream of LAMTOR-Rag, we generated *Tfeb* KO and *Lamtor1/Tfeb* DKO cells. Notably, similar to TFE3 depletion (Extended Data Figure 5d-g), loss of TFEB in either parental or *Lamtor1* KO cells abolished PRR-induced IFN responses (Extended Data Figure 5h-j). These findings demonstrate that TFEB/TFE3 may function as a positive regulator of IFN induction, independent of LAMTOR-Rag signaling.

In the revised manuscript, we omitted data involving overexpression of nuclear-targeted TFEB (S142A/S211A) to avoid potential misinterpretation (previously Extended Data Figure 10d-f). We speculate that this phosphorylation-deficient TFEB mutant may competitively inhibit IRF transcriptional activity in the nucleus during PRR signaling, thereby suppressing IFN expression. Future studies should explore the interplay between TFEB/TFE3 and IRFs at the transcriptional level, including potential shared regulatory motifs (e.g., CLEAR/ISRE elements) in IFN and ISG promoters.

Extended Data Figure 5

Extended Data Fig. 5 | Loss of TFEB/TFE3 impairs the PRR-induced IFN response.

d, Immunoblot analysis of TFEB and TFE3 in parental and *Lamtor1* KO, *Raga* KO, and *Ragc* KO cells using the same batch of samples as in Fig. 3g, with the same actin blot as the loading control.

e, Validation of TFE3 expression in *Tfe3* KO cells.

f, g, *Ifnb1* transcription (**f**) and IFN- β release (**g**) from *Tfe3* KO cells after PRR stimulation. Data are mean \pm SD, $n = 3$ biological replicates per group; two-way ANOVA followed by Tukey's test, adjusted P value as indicated.

h, Validation of TFEB expression in parental, *Lamtor1* KO, *Tfeb* KO, and *Lamtor1/Tfeb* DKO cells.

i, j, *Ifnb1* transcription (**i**) and IFN- β release (**j**) from the indicated cells after PRR stimulation. Data are mean \pm SD, $n = 3$ biological replicates per group; two-way ANOVA followed by Tukey's test, adjusted P value as indicated.

Regarding mTORC1's role in IFN induction, we have so far conducted comprehensive investigations using different experimental approaches including pharmacological inhibitors, Raptor deletion, and lysosomal targeting of Raptor in *Lamtor1* KO cells (**Extended Data Figure 4 in the revised manuscript**). Our data suggest that mTORC1—primarily an anabolic regulator—exerts its effects largely through protein translation, with the exception of TLR3-induced IFN transcription.

Additionally, we analyzed both canonical and non-canonical mTORC1 substrate phosphorylation in the case of LAMTOR-Rag depletion or loss of FLCN/DEPDC5 (see detailed response to specific comments). More importantly, we present new evidence that p38, acting downstream of LAMTOR-Rag, plays a critical role in the post-transcriptional regulation of IFN production (see Point 2; Figure 6g,h).

Finally, comparative analysis of IFN responses in *Depdc5* KO, *Mapk14/Mapk11* DKO, and *Depdc5/Mapk14/Mapk11* TKO cells revealed that, despite persistent mTORC1 hyperactivation, *Depdc5/Mapk14/Mapk11* TKO cells exhibited a significant impairment in IFN production (Reviewer Figure 2a-c). These findings highlight the critical importance of LAMTOR-Rag-p38 MAPK in controlling IFN output and strongly suggest that mTORC1 activation downstream of LAMTOR-Rag is insufficient to drive the IFN program.

Reviewer Figure 2

Reviewer Figure 2 | Signaling hierarchy of mTORC1 and p38 MAPK in regulating IFN induction.

a, Immunoblot analysis of S6K1, TFEB, and p38 MAPK phosphorylation in parental, *Mapk14/11* DKO, *Depdc5* KO, and *Depdc5/Mapk14/11* TKO cells.

b,c, *Ifnb1* transcription (**b**) and IFN-β release (**c**) from parental, *Mapk14/11* DKO, *Depdc5* KO, and *Depdc5/Mapk14/11* TKO cells after PRR stimulation. Data are mean ± SD, *n* = 3 biological replicates per group; two-way ANOVA followed by Tukey's test, adjusted *P* value as indicated.

In summary, we now have a clearer understanding of how lysosomal signaling is involved in regulating IFN induction during antiviral responses. The LAMTOR-Rag GTPase complex governs IFN- β production through dual transcriptional and post-transcriptional control. Mechanistically, Rag GTPase nucleotide cycling controls the IRF expression, priming the IFN transcriptional program. FLCN recruits p38 MAPK to lysosomes, where Rag nucleotide cycling dynamically activates lysosomal p38 MAPK, which is required to stabilize *Irf1* mRNA. Finally, nutrient cues drive Rag nucleotide cycling, thus coupling IFN production to cellular anabolic capacity.

Major comments

1) The authors provide data showing that p38 is present on lysosomal fractions and that its recruitment requires an intact LAMTOR-Rag hypercomplex, as well as the presence of active Rag dimers. However, the mechanistic underpinnings of its lysosomal localization and of its activation are less clear. More specifically:

a) Is the recruitment of p38 to lysosomes and its interaction to WT Rag dimers influenced by nutrients (amino acids, glucose) or by immune pathway activation (eg by poly(I:C), LPS, or R848)?

To examine the effects of nutrients on p38 lysosomal recruitment, we employed genetic models with opposing Rag GTPase loading states (*Flcn* KO versus *Depdc5* KO), as technical limitations precluded simultaneous amino acid manipulation and magnetic sphere uptake approaches.

Figure 6

Fig. 6 | p38 MAPK is required to regulate *Irf1* mRNA stability.

d, Immunoblot analysis of phagolysosomes isolated from parental, *Flcn* KO, and *Depdc5* KO cells. The asterisk (*) indicates a non-specific band detected by the FLCN antibody.

f, Representative micrographs showing co-localization of mBaoJin-MAPK14 with LysoView-positive lysosomes. Scale bars: 10 μ m.

Lyso-IP assays in these models demonstrated that: (1) *Flcn* KO cells (maintaining RagC in a constitutively GTP-bound state) exhibited complete loss of p38 phagolysosomal localization, while (2) *Depdc5* KO cells (with constitutively active Rag nucleotide loading) showed substantially enhanced p38 recruitment to phagolysosomes (**Figure 6d**). Supporting these observations, live-cell imaging showed abolished co-localization of mBaoJin-MAPK14 with LysoView-positive lysosomes in *Flcn* KO versus *Depdc5* KO or control cells (**Figure 6f**).

Additionally, Lyso-IP analysis in RAW cells revealed that PRR stimulation significantly enhances p38 MAPK recruitment to lysosomes (**Extended Data Figure 10h**).

Extended Data Figure 10

Extended Data Fig. 10 | p38 MAPK activation is critical for PRR-induced IFN induction.

h, Immunoblot analysis of phagolysosomes isolated from parental cells after PRR stimulation.

Together, these results establish that both PRR activation status and Rag GTPase nucleotide loading states serve as critical determinants of p38 MAPK lysosomal localization.

b) Whether this is a specific pool of p38 that localizes on lysosomes and is regulated specifically by lysosome-resident mechanisms, or this is the generic p38 MAPK signaling pathway is not clear. Does the activation of p38 by other means (eg osmolestress, oxidative stress) mimic or influence the immune response activation to induce IFN β expression? If yes, does this also happen in Rag/LAMTOR-deficient cells?

Unlike canonical MAPKs, which primarily transduce mitogenic signals, the p38 family responds to diverse environmental and genotoxic stressors. The functional outcomes of p38 activation are tightly regulated by spatiotemporal context and stimulus specificity. For instance, in response to DNA double-strand breaks, p38 translocates to the nucleus and promotes DNA repair by phosphorylating transcription factors such as ATF-2 and c-Jun [PMID: 33504982]. In contrast, during innate immune signaling, p38 remains predominantly cytoplasmic, with a subset recruited to lysosomes via FLCN, where Rag GTPase activity modulates its activation. These findings suggest that the local pool of p38 and its context-dependent interaction networks dictate its functional outcomes. We have incorporated this discussion into the revised manuscript.

Reviewer Figure 3

Reviewer Figure 3 | p38 MAPK activation by osmotic and oxidative stress fails to induce *Ifnb1* transcription.

a, Immunoblot analysis of p38 MAPK phosphorylation in parental and *Mapk14/11* DKO cells following 4 h of stimulation with sorbitol (0.5 M) or sodium arsenite (250 μ M).

b, *Ifnb1* mRNA levels in parental, *Mapk14/11* DKO, and *Lamtor1* KO cells treated with sorbitol, sodium arsenite, and LPS. Data are mean \pm SD, $n = 3$ biological replicates per group; two-way ANOVA followed by Tukey's test, adjusted P value as indicated.

To explore whether activation of p38 by other means could induce IFN- β , we compared the effects of osmotic stress (sorbitol, 0.5M for 4 h) and oxidative stress (sodium arsenite, 250 μ M for 4 h) with one of the PRR agonists (LPS). Notably, these stressors had minimal effects on *Irfb1* transcription and IFN- β secretion compared to innate immune signaling (**Reviewer Figure 3a,b**). We did not study prolonged treatments because treatment with these stressors for more than 4 h resulted in cell death.

c) The authors should complement the lysosome enrichment assays using confocal microscopy experiments and colocalization analysis of p38 with a lysosomal marker to confirm the lysosomal localization of (a fraction of?) p38 and to properly assess potential changes in p38 localization upon amino acid starvation or immune signaling activation.

From the outset of this study, we made extensive efforts to establish reliable immunofluorescence assays in RAW macrophages. However, despite rigorous optimization attempts, we consistently observed non-specific fluorescence signals that prevented reliable interpretation of target protein localization (**Reviewer Figure 4a-c**).

Reviewer Figure 4

Reviewer Figure 4 | Non-specific immunofluorescence signals of p38 MAPK in RAW macrophages.

a,b Immunofluorescence staining of p38 MAPK and lysosomal marker Lamp2 using the indicated antibodies. Scale bars: 10 μ m.

c, Immunofluorescence staining of p38 MAPK and mitochondrial marker Tom20. Scale bars: 10 μ m.

Specifically, for p38 MAPK, staining in both parental and *Mapk14/11* DKO cells using two distinct antibodies (Cell Signaling Technology #8690 and Abcam #ab170099, both Rabbit mAbs) revealed persistent non-specific signals in DKO cells, confirming lack of specificity (**Reviewer Figure 4a-c**).

Furthermore, while co-staining with the lysosomal marker Lamp2 (Santa Cruz Biotechnology #sc-18822, Mouse mAb) showed expected punctate distribution, parallel experiments with the mitochondrial marker Tom20 (Santa Cruz Biotechnology #sc-17764, Mouse mAb) also yielded similar staining patterns, strongly suggesting non-specific antibody cross-reactivity under our experimental conditions (**Reviewer Figure 4c**).

Reviewer Figure 5

Reviewer Figure 5 | Subcellular distribution of mBaoJin-MAPK14 in response to nutrient starvation. Live-cell imaging of mBaoJin-MAPK14 and LysoView-labeled lysosomes in RAW cells. Scale bars: 10 µm.

To overcome these technical limitations, we overexpressed mBaoJin-MAPK14 in RAW macrophages by transient transfection. Live-cell imaging revealed that mBaoJin-MAPK14 exhibited punctate distribution signals overlapping with LysoView-labeled lysosomes under nutrient-replete conditions. Notably, this punctate distribution pattern became largely diffuse following amino acid starvation (**Reviewer Figure 5**), suggesting dynamic regulation of p38 MAPK subcellular localization in response to metabolic stress.

d) If none of the upstream activating kinases (p38 MAPKKs) are present on lysosomes, why would the Rags and the p38 recruitment on lysosomes be required for its activation? Can the authors discuss potential explanations?

Indeed, the upstream activating kinases of p38 were barely detectable in the purified phagolysosomal fractions, suggesting an alternative regulatory mechanism. Our findings demonstrate that FLCN interacts with p38 MAPK and is required for its lysosomal recruitment. We propose that Rag GTPases mediate a non-canonical, lysosome-specific activation of p38 MAPK, distinct from the classical three-tiered MAPK signaling cascade. Based on our data, we hypothesize that the nucleotide states of Rag GTPases regulate localized p38 MAPK activation by recruiting potential modulators to spatiotemporally control p38 activity at lysosomes. However, the exact mechanistic details remain to be elucidated and warrant further investigation. We have incorporated this discussion into the revised manuscript.

e) Finally, the authors have not identified any of the downstream targets connecting p38 to the IFN response. While I agree that this is beyond the scope of this study, it would be beneficial that the authors expand the discussion to refer to potential mechanistic links.

Please see **Point 2**. Our new findings demonstrate a critical role for p38 in the post-transcriptional stabilization of *Irf1* mRNA. Regarding the mechanism by which p38 MAPK regulates the IFN program post-transcriptionally, prior studies have established that RNA-binding proteins HuR (ELAVL1) and Tristetraprolin (TTP/ZFP36) reciprocally modulate the stability of IFN and ISG mRNAs by binding to AU-rich elements (AREs) in their 3'-untranslated regions [PMID: 25678110, 34038724]. Specifically, the p38 substrate MK2 phosphorylates TTP, inhibiting its recruitment of the CCR4-NOT deadenylase complex and thereby stabilizing target mRNAs [PMID: 33504982, 21078877]. Conversely, during DNA damage, p38-mediated phosphorylation of HuR promotes its cytoplasmic translocation, enhancing mRNA stabilization to enforce cell-cycle arrest [PMID: 19528229]. Together, these observations suggest that p38 may coordinate IFN mRNA stability through opposing regulatory effects on TTP and HuR. However, whether this mechanism operates at lysosomes remains an open question. We have incorporated this discussion into the revised manuscript.

2) In ED Fig. 10, the authors investigate the potential involvement of the TFEB/TFE3 branch downstream of mTORC1 signaling in the regulation of the IFN response. Importantly, these data indicate that proper TFEB/TFE3 signaling is necessary for the IFN response.

Given that the lysosomal LAMTOR-Rag related machinery is primarily responsible for the regulation of the lysosomal, non-canonical mTORC1 substrates like TFEB/TFE3, and to a lesser extent for the phosphorylation of the canonical S6K and 4EBP1 (PMID: 31672913, 39385049) a plausible alternative interpretation of these findings is that the LAMTOR-Rag complex still regulates IFN expression via changes in mTORC1 activity that is directed towards the phosphorylation of its lysosomal substrates.

In support of this scenario, the authors observe no differences in canonical mTORC1 signaling upon loss or Lamtor1 expression or its lysosomal tethering.

Additional controls and a more detailed mechanistic analysis are required to convincingly exclude a potential role of mTORC1-TFEB/TFE3 signaling in the IFN-related downstream processes described in this manuscript.

Please see **Point 3 (Extended Data Figure 5a–g; Reviewer Figure 2a–c)**. To further validate the positive role of TFEB/TFE3 in regulating IFN induction and evaluate potential functional redundancy between these transcription factors, we generated *Flcn/Tfeb* DKO and *Tfeb/Tfe3* DKO cells using CRISPR (**Reviewer Figure 8a**). Depletion of TFEB in *Flcn* KO cells failed to rescue the IFN defects. Moreover, simultaneous depletion of both TFEB and TFE3 abolished *Ifnb1* transcription (**Reviewer Figure 8b**), consistent with the effects observed upon single deletions.

Reviewer Figure 8

a

b

Reviewer Figure 8 | TFEB/TFE3 positively regulates IFN induction.

a, Validation of TFEB/TFE3 expression in parental and *Flcn* KO, *Flcn/Tfeb* DKO, and *Tfeb/Tfe3* DKO cells.

b, *Ifnb1* mRNA levels in parental, *Flcn* KO, *Flcn/Tfeb* DKO, and *Tfeb/Tfe3* DKO cells following PRR stimulation. Data are mean \pm SD, $n = 3$ biological replicates per group; two-way ANOVA followed by Tukey's test, adjusted P value as indicated.

Together, these findings demonstrate that TFEB/TFE3 does not suppress IFN production downstream of the LAMTOR-Rag axis. Rather, they function as positive transcriptional regulators essential for IFN gene expression.

a) Fig. 1i and 4a: No difference in canonical mTORC1 signaling (S6K phosphorylation) is observed between Lamtor1 WT and 3A reconstituted cells. How does TFEB phosphorylation look in these cells?

Figure 1 (prev. Figure 1i)

prev. Figure 4

Fig. 1 | The lysosomal LAMTOR is antiviral and triggers the IFN signature via PRRs.

h, Innate immune signaling in WT and 3A cells in response to PRR stimulation. Cells were lysed and analyzed for the levels of the indicated proteins and phosphorylation status of S6K1 (T389), TFEB (S122), TBK1 (S172), IRF-3 (S396), and STING (S365).

TFEB phosphorylation levels in 3A-reconstituted cells were abolished under both basal and PRR-stimulated conditions (new Figure 1h and previously Figure 4a).

b) Fig. 4: The authors show that Torin treatments, that are expected to block the phosphorylation of all mTORC1 substrates (including TFEB and TEF3), do not block the transcriptional induction of IFNbeta. However, long-term mTORC1 inhibition may be causing additional, more generic effects (for instance, by affecting global translation) that would complicate the interpretation of the data.

Does blockage of lysosomal function (eg by using bafilomycin) that specifically affects the phosphorylation of TFEB/TFE3 without affecting mTORC1 activity also block the IFN response? If yes, is this rescued in cell expressing active RagA?

We examined the impact of Bafilomycin A1 on signaling events and IFN responses in RAW macrophages. Although Bafilomycin A1 reduced TFEB phosphorylation, it had minimal effect on *Irf1* transcription and IFN-β secretion compared to PRR stimulation, as exemplified by Poly(I:C) treatment (Reviewer Figure 6a,b).

Reviewer Figure 6

Reviewer Figure 6 | Bafilomycin A1 minimally affects IFN induction.

a, Immunoblot analysis of TFEB, S6K1, and p38 MAPK phosphorylation in cells treated with Bafilomycin A1 or Torin 1.

b, *Ifnb1* mRNA levels (left) and IFN- β release (right) from cells treated with Bafilomycin A1 or Poly(I:C). Data are mean \pm SD, $n = 3$ biological replicates per group; one-way ANOVA followed by Dunnett's test, adjusted P value as indicated.

c) In addition, is mTOR localization or Rag nucleotide loading affected upon immune signaling activation?

The protein levels of mTOR and Raptor remained largely unchanged in the phagolysosomal fractions of cells treated with PRR agonists. In contrast, PRR stimulation enhanced the presence of p38 MAPK in phagolysosomes (Extended Data Figure 10h).

Extended Data Figure 10

Extended Data Fig. 10 | p38 MAPK activation is critical for PRR-induced IFN induction.

h, Immunoblot analysis of phagolysosomes isolated from parental cells after PRR stimulation.

d) In general, controls to test the effect of various treatments or genetic perturbations in mTORC1 signaling are missing from several panels. For example, this is relevant for panels 3c-e and 5-c,d,g,h showing the Rag KO and reconstituted lines (using TFEB phosphorylation), or panels 4f-g showing the Raptor KO and reconstitution lines (using S6K phosphorylation as read-out). Similarly, control immunoblots to assess the effects of rapamycin and Torin treatments on mTORC1 activity (using S6K, 4EBP and TFEB phosphorylation as read-outs) should be included in Fig. 4.

As requested, we analyzed mTORC1 signaling in related experiments as follows:

- Control immunoblots evaluating mTORC1 signaling following LAMTOR-Rag depletion, along with the effects of chemical inhibitors on mTORC1 activity, are now included (Extended Data Figure 4a,d).

Extended Data Figure 4

Extended Data Fig. 4 | mTORC1 and LAMTOR-Rag regulate the IFN program via distinct mechanisms.

a, Immunoblot analysis of resting-state mTORC1 signaling in parental, *Lamtor1* KO, *Rraga* KO, and *Rragc* KO cells. Cells were lysed and analyzed for the levels of the indicated proteins and phosphorylation status of S6K1 (T389), 4E-BP1 (S65), and S6 (S235/236).

d, Immunoblot analysis of mTORC1 signaling in cells treated with mTORC1 inhibitors. Cells were lysed and analyzed for the levels of the indicated proteins and phosphorylation status of TFEB (S122), S6K1 (T389), 4E-BP1 (S65), and S6 (S235/236).

- Control immunoblots assessing the S6K phosphorylation in *Rptor* KO and *Lamtor1* KO cells stably expressing Raptor-Rheb15 are now included (**Extended Data Figure 4e,i**).

Extended Data Figure 4

(prev. Figure 4g)

Extended Data Fig. 4 | mTORC1 and LAMTOR-Rag regulate the IFN program via distinct mechanisms.

e, Immunoblot analysis of S6K1 phosphorylation in parental and *Rptor* KO cells following PRR stimulation.

i, Immunoblot analysis of S6K1 phosphorylation in parental, *Lamtor1* KO, and *Lamtor1* KO cells stably expressing Raptor-Rheb15.

- Control immunoblots assessing the TFEB phosphorylation in Rag KO and reconstituted lines are now included (**Extended Data Figure 5a-c**; **Extended Data Figure 8e-h**).

Extended Data Figure 5

Extended Data Fig. 5 | Loss of TFEB/TFE3 impairs the PRR-induced IFN response.

a–c, Immunoblot analysis of TFEB phosphorylation in parental, *Lamtor1* KO (**a**), *Rraga* KO (**b**), and *Rragc* KO cells (**c**) following PRR stimulation.

Extended Data Figure 8

Extended Data Fig. 8 | Rag GTPases couple nutrient availability to IFN induction.

e–h, Immunoblot analysis of TFEB phosphorylation in *Rraga/c* KO cells reconstituted with WT, GTP-bound (**e,g**), and GDP-bound (**f,h**) forms of mouse RagA/C in response to PRR stimulation.

For consistency, we examined mTORC1 signaling under PRR stimulated conditions following the experimental conditions shown in **Figure 1h** (mock vs. treatment with PRR agonists, 4 h). We have included these new data in the revised manuscript and have described them in the Result sections on mTORC1-TFEB/TFE3 signaling.

With all these controls in place, we removed the data regarding signaling analysis in LAMTOR1 WT and 3A reconstituted cells (**previously Figure 4a**).

e) Accordingly, microscopy analyses to assess mTOR localization to lysosomes would be important to functionally characterize the various LAMTOR/Rag-deficient models.

As with our observations regarding p38 MAPK staining (see response to **Major Comment 1c**; **Reviewer Figure 4**), we encountered similar challenges when attempting to assess lysosomal localization of endogenous mTOR via immunofluorescence. Our extensive optimization efforts consistently yielded non-specific staining patterns that precluded reliable interpretation of mTOR subcellular distribution (**Reviewer Figure 7**).

Reviewer Figure 7

Reviewer Figure 7 | Non-specific immunofluorescence signals of mTOR in RAW macrophages. Immunofluorescence staining of mTOR and lysosomal marker Lamp2. Scale bars: 10 μm .

f) ED Fig 10a: Is it the total protein levels or changes in the electrophoretic mobility of TFEB due to dephosphorylation that is observed in this panel?

When comparing *Lamtor1* KO, *Tfeb* KO, and *Lamtor1/Tfeb* DKO cells, loss of *Lamtor1* resulted in decreased total TFEB/TFE3 levels, which in turn led to decreased phosphorylation levels (now **Extended Data Figure 5d,e**). In general, we observed an overall decrease in total TFEB/TFE3 protein levels after LAMTOR-Rag disruption (see also **Extended Data Figure 5a-c**).

Extended Data Figure 5

Extended Data Fig. 5 | Loss of TFEB/TFE3 impairs the PRR-induced IFN response.

d, Immunoblot analysis of TFEB and TFE3 in parental, *Lamtor1* KO, *Rraga* KO, and *Rragc* KO cells using the same batch of samples as in Fig. 3g, with the same actin blot as the loading control.

e, Validation of TFE3 expression in *Tfeb* KO cells.

3) It is advisable to refer to the “Ragulator” complex as “LAMTOR complex” (or at least as “LAMTOR/Ragulator complex”) mainly for two reasons: 1) historical – this is the original name of this complex; 2) the exclusive use of the term ‘Ragulator’ can be misleading as it restricts the description of its function to the regulation of the Rags. This ignores completely additional molecular functions of this protein complex, such as the regulation of MAPK signaling on the lysosomal surface, as was previously demonstrated by the work of Lukas Huber and others.

On a related note, the authors here name the newly-identified LAMTOR-Rag-p38 axis as “Lysosomal Signaling For Interferon Induction” (or in short “LYSFII”) a practice that should be generally avoided. In particular, as neither the mechanism of p38 activation on lysosomes nor its downstream effectors that regulate IFN expression have been resolved in this study, the term “pathway” and giving it a special name is by no means justified.

We agree with your point of view regarding the terminology of the LAMTOR complex, as it more accurately reflects its broader functional roles beyond mTORC1 regulation. In the revised manuscript, we have adopted “LAMTOR complex” consistently to maintain clarity and precision. Furthermore, we now define the

LAMTOR-Rag-p38 MAPK axis as the key lysosomal signaling module essential for IFN production, thereby avoiding unnecessary or ambiguous terminology.

Minor comments

- It may be useful to a broader readership to also briefly describe the principles of lysosomal mTORC1 signaling and introduce key complexes and mechanisms in the introduction?

We have included this aspect in the Introduction of the revised manuscript.

- Lines 162-163: The authors refer to “RagA, RagB, and RagC isoforms” which is not correct, as the different Rags are not isoforms of the same gene, but rather encoded by 4 distinct paralogous genes.

Thanks for pointing this out. We have corrected it in the revised manuscript.

Reviewer #2 (Remarks to the Author):

In this manuscript, Feng et al. describe a new role for the Ragulator-Rag GTPase complex in the antiviral response. They found that Ragulator/Rag GTPase activation is required for IFN-beta transcription, via a mechanism that seems to be independent from mTORC1 and IRF3 activity, and dependent on p38 activation.

While the manuscript contains potentially impactful discoveries and presents interesting observations, several critical issues remain unresolved, limiting its mechanistic depth and clarity. The work is predominantly descriptive, and key mechanistic aspects of how the Ragulator-Rag GTPase complex modulates IFN transcription remain unexplored.

The authors propose that p38 is a central effector in this pathway, yet how p38 is recruited to the lysosomal membrane, activated, and connected to the observed transcriptional outcomes is not addressed.

Furthermore, some aspects of the proposed model appear inconsistent with the provided data (discussed below). More importantly, the role of p38 in modulating IFN transcription is insufficiently characterized. Despite the emphasis on its relevance, the mechanism by which p38 contributes to the proposed “IRF3-independent” regulation of IFN-beta transcription remains unknown.

Puzzlingly, the authors propose in the discussion a potential involvement of the MiT-TFE transcription factors TFEB and TFE3, which are well-established effectors of the Rag GTPases and non-canonical mTORC1 substrates, providing data that support a possible role of these proteins in IFN transcription. However, the authors did not investigate this further to substantiate their involvement (see below).

In summary, while the findings reported here are interesting and may expand our understanding of lysosomal signaling in antiviral responses, the impact of this manuscript is limited by several significant mechanistic gaps.

Dear Reviewer #2,

Thank you very much for recognizing the potentially impactful findings in our work and for providing thoughtful suggestions to advance the mechanistic understanding of lysosomal signaling in IFN regulation.

In the revised manuscript, we demonstrate a regulatory role for p38 MAPK, specifically in the post-transcriptional stabilization of *Ifnb1* mRNA. During innate immune activation, while p38 predominantly localizes to the cytoplasm, a subset is recruited to lysosomes via FLCN, where Rag GTPase activity regulates its activation. Importantly, genetic ablation of *Mapk14/Mapk11* abolishes IFN responses and increases viral susceptibility even under active Rag signaling, establishing its essential role in linking lysosomal signaling to *Ifnb1* transcript longevity.

To further strengthen the mechanistic framework, we have:

- Systematically analyzed the epistasis between p38 MAPK and LAMTOR-Rag by comparing *Fln* versus *Depdc5* deletion
- Examined ectopic expression of active Rags in *Mapk14/Mapk11* DKO cells
- Investigated *Depdc5* deletion in *Mapk14/Mapk11* DKO cells

Furthermore, we present new evidence clarifying TFEB/TFE3 as positive transcriptional regulators of IFN that function independently of the LAMTOR-Rag axis.

Our integrated findings reveal that lysosomal signaling coordinates IFN induction through the following interconnected mechanisms:

1. Transcriptional control via LAMTOR-Rag-mediated regulation of IRF expression;
2. Post-transcriptional regulation through FLCN-dependent recruitment and Rag-mediated activation of p38 MAPK, which stabilizes *Ifnb1* mRNA;
3. Metabolic coupling whereby nutrient-sensitive Rag nucleotide cycling links IFN production to cellular anabolic state.

These results position the lysosome as a central regulatory hub that integrates metabolic and immune signaling to control antiviral interferon responses.

Detailed point-by-point responses follow below.

Specific points:

1. No mechanism for how the Ragulator/RagGTPase complex modulates IFN-beta transcription is provided. The authors implicate p38 as a key mediator of this pathway, but how p38 functions in this context remains unexplored, and no evidence regarding its specific targets in the context of IFN-beta transcription has been provided.

Does p38 phosphorylate a specific transcription factor, co-factor, or chromatin-associated protein that drives IFN-beta transcription independently of IRF3? Identifying such targets is essential to support the proposed mechanism.

Furthermore, the model presented in Figure 8h suggests that p38 is recruited to the lysosomal membrane by the Ragulator/Rag GTPase complex, where it becomes activated and modulates IFN- β transcription. However, this model conflicts with the data.

Specifically, the authors show that p38 specifically binds to inactive GTP-RagC (Figure 8g). Yet, they show that GTP-bound RagC cannot support IFN production, whereas wild-type RagC or GDP-bound RagC can fully rescue IFN production in RagC-KO cells (Figure 5f).

Importantly, p38 activation is impaired in RagC-KO cells (Figure 7g), thus also excluding a possible mechanism whereby GTP-RagC sequesters and inhibits p38. These contradictions undermine the validity of the proposed model.

In the revised manuscript, we have substantially strengthened the evidence for epistasis between p38 MAPK and LAMTOR-Rag, demonstrated a critical role for p38 in regulating *Ifnb1* mRNA stability, and elucidated the molecular mechanisms underlying p38 recruitment to lysosomes.

1. Strengthening the epistatic relationship between p38 MAPK and LAMTOR-Rag and elucidating the mechanisms of p38 lysosomal recruitment

To delineate the epistasis between p38 MAPK and LAMTOR-Rag, we generated *Fln* KO and *Depdc5* KO cells, which exhibit opposing Rag GTPase nucleotide states. Consistent with the loss of LAMTOR-Rag,

FLCN depletion abolished p38 MAPK phosphorylation and the IFN response. Conversely, DEPDC5-deficient cells displayed enhanced p38 phosphorylation and a stronger IFN response (**new Figure 5a-c**).

Crucially, ectopic expression of RagA-GTP or RagC-GDP in *Mapk14/Mapk11* DKO cells did not restore the IFN program (**Figure 5d-g**). These findings strongly suggest that p38 MAPK functions downstream of LAMTOR-Rag GTPases to regulate IFN induction.

Figure 5

Fig. 5 | Rag nucleotide-loading state controls p38 MAPK activation necessary for IFN induction.

a, Immunoblot analysis of p38 MAPK phosphorylation in *Rragc* KO, *Flcn* KO, and *Depdc5* KO cells compared with parental cells.

b, *lfnb1* mRNA expression in parental, *Flcn* KO, and *Depdc5* KO cells after PRR stimulation. Data are mean \pm SD, n = 3 biological replicates per group; two-way ANOVA followed by Tukey's test, adjusted P value as indicated.

c, IFN- β secretion from parental, *Flcn* KO, and *Depdc5* KO cells after PRR stimulation, as measured by ELISA. Data are mean \pm SD, n = 3 biological replicates per group; two-way ANOVA followed by Tukey's test, adjusted P value as indicated.

d, Ectopic expression of FLAG-tagged RagA-GTP (Q66L) or RagC-GDP (S74N) in *Mapk14/11* DKO cells.

e, *Ifnb1* mRNA expression in *Mapk14/11* DKO cells expressing FLAG-tagged RagA-GTP or RagC-GDP after PRR stimulation. Data are mean \pm SD, n = 3 biological replicates per group; two-way ANOVA followed by Tukey's test, adjusted P value as indicated.

f, IFN- β secretion from *Mapk14/11* DKO cells expressing FLAG-tagged RagA-GTP or RagC-GDP after PRR stimulation. Data are mean \pm SD, n = 3 biological replicates per group; two-way ANOVA followed by Tukey's test, adjusted P value as indicated.

g, Schematic showing epistasis across the GATOR1 and FLCN-FNIP1/2 complex, RagA/C GTPases, and p38 MAPK.

Since the distinct Rag GTPase nucleotide states specifically affected p38 MAPK phosphorylation but not its downstream nuclear targets ATF-2 and c-Jun (**Figure 4j in the revised manuscript**), we examined p38 MAPK subcellular localization following PRR stimulation. Nuclear-cytoplasmic fractionation assays revealed predominant cytoplasmic localization of p38 MAPK, regardless of PRR stimulation or *Lamtor1* status (**new Figure 6a**). Using polymer-based magnetic bead isolation, we demonstrated the presence of p38 MAPK in phagolysosomal fractions (**Figure 6b,c**).

Figure 6

Fig. 6 | p38 MAPK is required to regulate *Ifnb1* mRNA stability.

a, Nuclear-cytoplasmic fractionation assays to determine the p38 MAPK localization in response to PRR stimulation.

b, Schematic workflow for the isolation of native phagolysosomes.

c, Immunoblot analysis of phagolysosomes isolated from parental and *Lamtor1* KO cells. WCL, whole cell lysate.

d, Immunoblot analysis of phagolysosomes isolated from parental, *Flcn* KO, and *Depdc5* KO cells. The asterisk (*) indicates a non-specific band detected by the FLCN antibody.

e, Interaction between mBaoJin-MAPK14 and FLAG-tagged mouse FLCN or the indicated variants, as assessed by FLAG immunoprecipitation in HEK293T cells.

f, Representative micrographs showing co-localization of mBaoJin-MAPK14 with LysoView-positive lysosomes. Scale bars: 10 μ m.

Building on our findings that FLCN and DEPDC5 oppositely regulate p38 phosphorylation and IFN responses (**Figure 5a–c**), we tested whether Rag GTPase nucleotide states control p38 lysosomal recruitment. *Depdc5* knockout markedly increased phagolysosomal localization of both Raptor and p38, while *Fln* knockout abolished p38 phagolysosomal association (**Figure 6d**). Co-immunoprecipitation demonstrated a specific interaction between p38 MAPK and FLCN, requiring both Longin and DENN domains (**Figure 6e**). Supporting these observations, live-cell imaging revealed abolished co-localization of mBaoJin-MAPK14 with LysoView-positive lysosomes in *Fln* KO versus *Depdc5* KO or control cells (**Figure 6f**).

Collectively, these data identify FLCN as a key mediator of p38 MAPK lysosomal recruitment, directly coupling Rag GTPase nucleotide status to spatially regulated p38 activation.

2. Demonstrating the specific role of p38 in IFN induction, which is to regulate *Ifnb1* mRNA transcript stability.

To corroborate the key role of p38 in IFN induction, we depleted *Mapk14* and/or *Mapk11* from cells using CRISPR. Loss of either or both isoforms impaired *Ifnb1* transcription and IFN- β secretion (**Extended Data Figure 10a,b; Figure 5e,f**), establishing p38's involvement in regulating IFN responses. Based on these findings, we have removed the p38 MAPK chemical inhibition data (**previously Figure 8a-d and Extended data Figure 9d-f**) from the revised manuscript.

Extended Data Figure 10

Extended Data Fig. 10 | p38 MAPK activation is critical for PRR-induced IFN induction.

a,b, *Ifnb1* transcription (**a**) and IFN- β release (**b**) from parental, *Mapk14* KO, and *Mapk11* KO cells after PRR stimulation. Data are mean \pm SD, $n = 3$ biological replicates per group; two-way ANOVA followed by Tukey's test, adjusted P value as indicated.

We then assessed whether p38 kinase activity was required for this regulation. Overexpression of a kinase-dead p38 mutant (D168A) significantly suppressed IFN responses, whereas a constitutively active mutant (D176A/F327S) did not further enhance responses beyond wild-type levels (**Extended Data Figure 10f,g**). These data suggest p38 activity is necessary but not limiting.

Extended Data Figure 10

Extended Data Fig. 10 | p38 MAPK activation is critical for PRR-induced IFN induction.

f,g, *lfnb1* transcription (**f**) and IFN- β release (**g**) from parental cells or cells expressing MAPK14 WT, D168A, or D176A/F327S after PRR stimulation. Data are mean \pm SD, $n = 3$ biological replicates per group; two-way ANOVA followed by Tukey's test, adjusted P value as indicated.

Considering the inherent instability of *lfnb1* mRNA, we hypothesized that extranuclear p38 MAPK activity during PRR stimulation might regulate its stability. We employed Roadblock-qPCR to measure the half-life of *lfnb1* mRNA. Remarkably, *lfnb1* mRNA degradation was significantly accelerated in both *Lamtor1* KO and *Mapk14/Mapk11* DKO cells relative to parental cells (**Figure 6g**). In contrast, DEPDC5 deficiency substantially prolonged *lfnb1* mRNA half-life, an effect that was reversed by *Mapk14/Mapk11* deletion (**Figure 6h**). These results demonstrate that p38 MAPK critically regulates *lfnb1* mRNA stability downstream of the LAMTOR-Rag GTPases.

Figure 6

Fig. 6 | p38 MAPK is required to regulate *Ifnb1* mRNA stability.

g, Measurements of *Ifnb1* mRNA decay using Roadblock-qPCR in parental, *Lamtor1* KO, and *Mapk14/11* DKO cells.

h, Measurements of *Ifnb1* mRNA decay using Roadblock-qPCR in parental, *Depdc5* KO, *Mapk14/11* DKO, and *Depdc5/Mapk14/Mapk11* TKO cells.

i, Representative micrographs of RAW macrophages of the indicated genotypes following infection with IAV PR8-NS1-GFP at an MOI of 0.25 for 24 h. Scale bars: 100 μ m.

Consistent with this post-transcriptional role of p38, impaired IFN production in *Lamtor1* KO, *Ficn* KO, and *Mapk14/11* DKO cells increased viral susceptibility, whereas the potent antiviral state in *Depdc5* KO cells was strictly dependent on p38 MAPK activity (**Figure 6i**).

How does p38 MAPK regulate the *Ifnb1* mRNA stability? Prior studies have established that RNA-binding proteins HuR (ELAVL1) and Tristetraprolin (TTP/ZFP36) reciprocally modulate the stability of IFN and ISG mRNAs by binding to AU-rich elements (AREs) in their 3'-untranslated regions [PMID: 25678110, 34038724]. Specifically, the p38 substrate MK2 phosphorylates TTP, inhibiting its recruitment of the CCR4-NOT deadenylase complex and thereby stabilizing target mRNAs [PMID: 33504982, 21078877]. Conversely, during DNA damage, p38-mediated phosphorylation of HuR promotes its cytoplasmic translocation, enhancing mRNA stabilization to enforce cell-cycle arrest [PMID: 19528229]. Together, these observations suggest that p38 may coordinate IFN mRNA stability through opposing regulatory effects on TTP and HuR. However, whether this mechanism operates at lysosomes remains an open question. We have incorporated this discussion into the revised manuscript.

Finally, comparative analysis of IFN responses in *Depdc5* KO, *Mapk14/Mapk11* DKO, and *Depdc5/Mapk14/Mapk11* TKO cells revealed that, despite persistent mTORC1 hyperactivation, *Depdc5/Mapk14/Mapk11* TKO cells exhibited a significant impairment in IFN production (**Reviewer Figure 2a-c**). These findings highlight the critical importance of LAMTOR-Rag-p38 MAPK in controlling IFN output and strongly suggest that mTORC1 activation downstream of LAMTOR-Rag is insufficient to drive the IFN program.

Reviewer Figure 2

Reviewer Figure 2 | Signaling hierarchy of mTORC1 and p38 MAPK in regulating IFN induction.

a, Immunoblot analysis of S6K1, TFEB, and p38 MAPK phosphorylation in parental, *Mapk14/11* DKO, *Depdc5* KO, and *Depdc5/Mapk14/11* TKO cells.

b,c, *lfnb1* transcription (**b**) and IFN- β release (**c**) from parental, *Mapk14/11* DKO, *Depdc5* KO, and *Depdc5/Mapk14/11* TKO cells after PRR stimulation. Data are mean \pm SD, $n = 3$ biological replicates per group; two-way ANOVA followed by Tukey's test, adjusted P value as indicated.

In summary, we now have a clearer understanding of how lysosomal signaling is involved in regulating IFN induction during antiviral responses. The LAMTOR-Rag GTPase complex governs IFN- β production through dual transcriptional and post-transcriptional control. Mechanistically, Rag GTPase nucleotide cycling controls the IRF expression, priming the IFN transcriptional program. FLCN recruits p38 MAPK to lysosomes, where Rag nucleotide cycling dynamically activates lysosomal p38 MAPK, which is required to stabilize *lfnb1* mRNA. Finally, nutrient cues drive Rag nucleotide cycling, thus coupling IFN production to cellular anabolic capacity.

2. The manuscript suggests a potential involvement of MiT/TFE transcription factors (TFEB and TFE3) in the regulation of IFN production, yet this critical aspect has not been examined in sufficient detail.

TFEB and TFE3 are well known non-canonical mTORC1 substrates and their phosphorylation is strictly dependent on the Regulator/Rag GTPase axis, via a mechanism that is different from the one responsible

for the phosphorylation of canonical mTORC1 substrates S6K and 4E-BP1. In LAMTOR1- or Rag-deficient cells, the absence of this phosphorylation leads to TFEB/TFE3 constitutive activity and nuclear localization.

Results shown in Extended Data 10 highlight that overexpression of a constitutively active TFEB mutant is sufficient to suppress IFN-beta production, strongly implicating MiT/TFE factors in the observed phenotype. Despite this, the authors did not experimentally investigate the function of these transcription factors in this context.

To determine the role of MiT/TFE factors in regulating IFN production, the authors should test whether siRNA-mediated silencing of both TFEB and TFE3 in LAMTOR1-, RagA-, and RagC-KO cells rescues IFN-beta production. Given that both factors can act redundantly, simultaneous silencing is essential to avoid compensatory effects.

In addition, IFN transcription and secretion should also be assessed in folliculin (FLCN)-deficient cells, which exhibit constitutive MiT/TFE activation due to impaired RagC/D GTPase activity. The role of MiT/TFE factors should also be tested in this context via siRNA-mediated depletion.

The lysosomal LAMTOR-Rag-FLCN axis is known to negatively regulate TFEB/TFE3 nuclear translocation and their transcriptional activities. Thus, disruption of this axis would be expected to induce constitutive TFEB/TFE3 activation, which might contribute to the observed impairment of IFN induction. Indeed, depletion of either *Lamtor1*, *RagA* or *RagC* abolished the TFEB phosphorylation (**Extended Data Figure 5a-c**).

To investigate whether TFEB/TFE3 suppresses IFN transcription downstream of LAMTOR-Rag, we generated *Tfeb* KO and *Lamtor1/Tfeb* DKO cells. However, similar to TFE3 depletion (**Extended Data Figure 5d-g**), loss of TFEB in either parental or *Lamtor1* KO cells abolished PRR-induced IFN responses (**Extended Data Figure 5h-j**). These findings demonstrate that TFEB/TFE3 paradoxically acts as positive regulators of IFN induction, independent of LAMTOR-Rag signaling.

Extended Data Figure 5

Extended Data Fig. 5 | Loss of TFEB/TFE3 impairs the PRR-induced IFN response.

a-c, Immunoblot analysis of TFEB phosphorylation in parental, *Lamtor1* KO, *Rraga* KO, and *Rragc* KO cells following PRR stimulation.

d, Immunoblot analysis of TFEB and TFE3 in parental, *Lamtor1* KO, *Rraga* KO, and *Rragc* KO cells using the same batch of samples as in Fig. 3g, with the same actin blot as the loading control.

e, Validation of TFE3 expression in *Tfe3* KO cells.

f,g, *ifnb1* transcription (**f**) and IFN- β release (**g**) from *Tfe3* KO cells after PRR stimulation. Data are mean \pm SD, $n = 3$ biological replicates per group; two-way ANOVA followed by Tukey's test, adjusted P value as indicated.

h, Validation of TFEB expression in parental, *Lamtor1* KO, *Tfeb* KO, and *Lamtor1/Tfeb* DKO cells.

i,j, *ifnb1* transcription (**i**) and IFN- β release (**j**) from the indicated cells after PRR stimulation. Data are mean \pm SD, $n = 3$ biological replicates per group; two-way ANOVA followed by Tukey's test, adjusted P value as indicated.

To further validate the positive role of TFEB/TFE3 in regulating IFN induction and evaluate potential functional redundancy between these transcription factors, we generated *Flcn/Tfeb* DKO and *Tfeb/Tfe3* DKO cells using CRISPR (**Reviewer Figure 8a**). Depletion of TFEB in *Flcn* KO cells failed to rescue the IFN defects. Moreover, simultaneous depletion of both TFEB and TFE3 abolished *ifnb1* transcription (**Reviewer Figure 8b**), consistent with the effects observed upon single deletions.

Reviewer Figure 8

a

b

Reviewer Figure 8 | TFEB/TFE3 positively regulates IFN induction.

a, Validation of TFEB/TFE3 expression in parental and *Flcn* KO, *Flcn/Tfeb* DKO, and *Tfeb/Tfe3* DKO cells.

b, *ifnb1* mRNA levels in parental, *Flcn* KO, *Flcn/Tfeb* DKO, and *Tfeb/Tfe3* DKO cells following PRR stimulation. Data are mean \pm SD, $n = 3$ biological replicates per group; two-way ANOVA followed by Tukey's test, adjusted P value as indicated.

Together, these findings demonstrate that TFEB/TFE3 does not suppress IFN production downstream of the LAMTOR-Rag axis. Rather, they function as positive transcriptional regulators essential for IFN gene expression.

In the revised manuscript, we omitted data involving overexpression of nuclear-targeted TFEB (S142A/S211A) to avoid potential misinterpretation (**previously Extended Data Figure 10d-f**). We speculate that this phosphorylation-deficient TFEB mutant may competitively inhibit IRF transcriptional activity in the nucleus during PRR signaling, thereby suppressing IFN expression. Future studies should explore the interplay between TFEB/TFE3 and IRFs at the transcriptional level, including potential shared regulatory motifs (e.g., CLEAR/ISRE elements) in IFN and ISG promoters.

3. If the role of TFEB/TFE3 in suppressing IFN production is confirmed, the authors should investigate on

the mechanisms by which these transcription factors exert their function in this context. How do TFEB and TFE3 suppress IFN response?

Which MiT/TFE-dependent genes are responsible for IFN suppression? And how does this response relate with IRF3, which seems unable to mediate IFN transcription in Ragulator/Rag GTPase-deficient cells?

See response to Q2.

4. In Figure 4, the authors show that short-term mTORC1 inhibition using rapamycin and torin has no measurable effects on IFN-beta transcription. Based on this, they conclude that the effects observed in LAMTOR- and RagA/C-KO cells are "mTORC1-independent." However, this conclusion may be incorrect, as the experimental design relies exclusively on acute (30-minute) drug treatments, which are unlikely to fully capture the downstream effects of mTORC1 inhibition.

Indeed, if the suppression of IFN-beta transcription depends on TFEB/TFE3-induced gene expression, acute treatments may not provide sufficient duration to observe these transcription-dependent effects. Accordingly, the authors show that prolonged amino acid starvation (14 hours), a condition that activates TFEB, can phenocopy the effects of LAMTOR-KO, leading to reduced IFN- β production (Extended Data Fig 4).

This observation strongly suggests that the suppression of IFN transcription may be mediated, at least in part, by TFEB activation. To robustly test the role of mTORC1 inhibition in regulating IFN transcription, the authors should extend rapamycin and torin treatments to 24-48 hours and assess changes in IFN transcription and secretion following long-term treatments.

The activation status of TFEB (e.g., nuclear localization and target gene expression) after long-term mTORC1 inhibition and the effect of TFEB/TFE3 depletion in this context should also be assessed to determine their involvement.

In experiments involving mTORC1 inhibitors, the drugs were administered 30 minutes prior to PRR stimulation and maintained throughout the 14-hour time course. This methodological detail has now been explicitly stated in the figure legends of **Extended Data Figure 4b** in the revised manuscript.

Although prolonged amino acid starvation completely suppressed IFN transcription in response to Poly(I:C), R848, CpG-B, and LPS, it robustly enhanced IFN transcription following cGAMP stimulation instead (**Extended Data Figure 8I**). In alignment with this observation, recent studies have implicated that TFEB/TFE3 serve as direct downstream effectors of STING as part of the lysosomal quality control pathways [PMID: 40185098, 39689715, 39423796].

As detailed in our response to Q2, these findings (**Extended Data Figure 5a-j; Reviewer Figure 8a,b**) collectively demonstrate that TFEB/TFE3 positively regulates IFN transcription through a mechanism independent of the LAMTOR-Rag axis.

5. In Figure 4 d-e, the authors show that Raptor depletion strongly increases IFN-beta transcription and secretion. These data are in contrast with the authors' observations upon torin and rapamycin treatment, which does not affect IFN transcription while repressing its secretion (Figure 4 b-c). How do the authors explain these discrepancies?

The apparent discrepancy between Raptor depletion and pharmacological mTORC1 inhibition can be explained by their fundamentally different mechanisms of action. While Torin/rapamycin acutely inhibit mTORC1 kinase activity (and therefore protein translation and secretion), genetic Raptor ablation leads to complete mTORC1 complex disruption, which may induce compensatory signaling rewiring and broader transcriptional reprogramming.

6. The results of the shRNA screening presented in Supplementary Table 1 reveal that only a small subset of genes appear to have no effect on viral infectivity. This observation is puzzling, as it suggests a broad

influence of the targeted genes on viral replication. A critical question is whether this effect reflects a more general disruption of lysosomal function, given the lysosome essential role in antiviral defense mechanisms.

This should be assessed experimentally, by testing several parameters of lysosomal function and integrity including pH (e.g. lysosensor), hydrolyse activity (e.g. Magic Red), and lysosomal permeabilization (e.g. galectin3 and CHMP4 staining) in LAMTOR- and RagA/C-KO cells. Furthermore, the authors should assess whether pharmacological inhibition of lysosomal function via Bafylomycin or chloroquine treatment also affects IFN transcription.

Our shRNA screen specifically targeted 50 lysosome-localized proteins involved in signal transduction. Among these, 20 candidates showed significant effects on IAV infection, with knockdown resulting in >1.5-fold change in IAV-GFP fluorescence.

Notably, *Lamtor1* KO, *Rraga* KO, and *Rragc* KO macrophages maintained normal cellular morphology and growth characteristics throughout our experiments. Importantly, LysoSensor Yellow/Blue DND-160 measurements confirmed that deletion of these genes did not impair endolysosomal acidification (**Extended Data Fig. 3h**), demonstrating intact lysosomal pH regulation in these knockout cells.

Extended Data Figure 3

Extended Data Fig. 3 | LAMTOR-Rag interactions are crucial for IFN induction.

h, Measurement of endolysosomal pH in parental, *Lamtor1* KO, *Rraga* KO, and *Rragc* KO cells using LysoSensor™ Yellow/Blue DND-160. Data are mean ± SD, *n* = 3 biological replicates per group; one-way ANOVA followed by Dunnett's test, adjusted *P* value as indicated.

Furthermore, transcriptomic analysis of LAMTOR1 WT versus 3A mutant cells under PRR stimulation revealed specific cytokine regulation patterns: while IFNs and IFN-regulated cytokines were significantly downregulated, C-C motif chemokines and TNF ligand superfamily members showed marked upregulation (**Extended Data Fig. 2g**). This selective modulation indicates that LAMTOR-Rag signaling specifically governs IFN responses rather than globally impairing cytokine induction, highlighting its precise role in antiviral immunity.

Extended Data Figure 2

Extended Data Fig. 2 | Loss of LAMTOR abolishes the IFN signature.

g, Heatmap comparing the gene expression profiles of cytokines between LAMTOR1 WT and 3A cells in response to PRR stimulation.

Additionally, we examined the impact of Bafilomycin A1 on signaling events and IFN responses in RAW macrophages. Although Bafilomycin A1 reduced TFEB phosphorylation, it had minimal effect on *Irfb1* transcription and IFN- β secretion compared to PRR stimulation, as exemplified by Poly(I:C) treatment (Reviewer Figure 6a,b).

Reviewer Figure 6

Reviewer Figure 6 | Bafilomycin A1 minimally affects IFN induction.

a, Immunoblot analysis of TFEB, S6K1, and p38 MAPK phosphorylation in cells treated with Bafilomycin A1 or Torin 1.

b, *Irfb1* mRNA levels (left) and IFN- β release (right) from cells treated with Bafilomycin A1 or Poly(I:C). Data are mean \pm SD, $n = 3$ biological replicates per group; one-way ANOVA followed by Dunnett's test, adjusted P value as indicated.

REVIEWER #3 - Comments to the authors:

The authors of the study explore the role of the Ragulator-Rag GTPase complex in the regulation of interferon (IFN). They propose that the transcription of IFN beta is independent of the mTORC pathway and recognition through pattern recognition receptors (PRRs), leading them to name this pathway as Lysosomal Signaling For Interferon Induction (LYSFII). Their data supports some of their conclusions, making this an interesting observation.

However, the data presented does not completely rule out the metabolic pathways that could affect transcription. Therefore, this reviewer suggests that the paper be revised to tone down its broad implications and to consider alternative hypotheses more thoroughly. This would provide a more balanced and accurate representation of the findings and their significance within the broader context of immune regulation and metabolic pathways.

Dear Reviewer #3,

We sincerely appreciate your thoughtful evaluation of our manuscript and your emphasis on contextualizing our findings within the broader literature on lysosomal signaling and immunity.

In the revised manuscript, we demonstrate a regulatory role for p38 MAPK, specifically in the post-transcriptional stabilization of *Irfb1* mRNA. During innate immune activation, while p38 predominantly

localizes to the cytoplasm, a subset is recruited to lysosomes via FLCN, where Rag GTPase activity regulates its activation. Importantly, genetic ablation of *Mapk14/Mapk11* abolishes IFN responses and increases viral susceptibility even under active Rag signaling, establishing its essential role in linking lysosomal signaling to *Ifnb1* transcript longevity.

Our integrated findings reveal that lysosomal signaling coordinates IFN induction through the following interconnected mechanisms:

1. Transcriptional control via LAMTOR-Rag-mediated regulation of IRF expression;
2. Post-transcriptional regulation through FLCN-dependent recruitment and Rag-mediated activation of p38 MAPK, which stabilizes *Ifnb1* mRNA;
3. Metabolic coupling whereby nutrient-sensitive Rag nucleotide cycling links IFN production to cellular anabolic state.

These results position the lysosome as a central regulatory hub that integrates metabolic and immune signaling to control antiviral interferon responses.

Below, we address the major comments with new data and clarifications incorporated into the revised manuscript.

Major comments

The Ragulator-Rag GTPase complex has been previously implicated in the induction of pro-inflammatory cytokines. This raises the question of whether there is a general increase in the inflammatory cytokine response, not restricted solely to IFN. This needs to be experimentally tested. If this is the case, it would be inappropriate to name this pathway LYSFII. If the response involves a general induction of cytokines, it is difficult to imagine that metabolic pathways are not perturbed. Although the authors conduct some experiments to demonstrate that the mTORC pathway is not involved, these experiments do not entirely rule out the dysregulation of mTORC leading to modulation of cytokine expression.

Clear experimental evidence is needed to show that alternative transcription factors activate IFNs. The paper primarily focuses on IFN and seems to overlook the existing literature on the function of the Ragulator-Rag complex in immunity. The experiments are designed in such a way that they do not address alternative hypotheses.

1. Specificity of the IFN response vs. broad cytokine regulation

You raised an important question about whether the LAMTOR-Rag complex broadly regulates inflammatory cytokines beyond IFN- β . To address this, we analyzed cytokine expression profiles in LAMTOR1 3A versus WT macrophages under PRR stimulation using RNA-seq data. While IFN- β and IFN-regulated genes (e.g., *Cxcl10*, *Isg15*) were significantly downregulated, other cytokines (e.g., *Tnf*, *Il6*, *Ccl2/4*) were either unaffected or upregulated (**Extended Data Fig. 2g**).

Extended Data Figure 2

Extended Data Fig. 2 | Loss of LAMTOR abolishes the IFN signature.

g, Heatmap comparing the gene expression profiles of cytokines between LAMTOR1 WT and 3A cells in response to PRR stimulation.

In support of this analysis, qRT-PCR analysis confirmed the transcriptional upregulation of *Ccl2* and *Ccl4* following disruption of LAMTOR-Rag upon LPS stimulation (**Reviewer Figure 9**).

Figure for reviewers removed

This selective modulation argues against a generalized inflammatory defect and supports a specific role for LAMTOR-Rag in IFN induction.

We agree that the name "LYSFII" overemphasized the pathway's exclusivity to IFN. In the revised manuscript, we refer to this axis as the "**LAMTOR-Rag-p38 MAPK module**" to better reflect its mechanistic role in IFN regulation while acknowledging potential crosstalk with other immune-metabolic pathways.

2. Metabolic contributions to IFN regulation

We acknowledge that loss of LAMTOR-Rag could indirectly affect IFN through metabolic perturbations. However, several lines of evidence argue against this being the primary driver:

- **mTORC1-independent effects:**

Prolonged mTORC1 inhibition using rapamycin/Torin (**Extended Data Fig. 4b-d**) or Raptor depletion (**Extended Data Fig. 4e-h**) did not fully recapitulate the IFN transcriptional suppression seen in *Lamtor1* KO cells (**Extended Data Fig. 4f,g**). Furthermore, lysosomal targeting of Raptor, which induces mTORC1 constitutive activation, failed to rescue the IFN defect caused by *Lamtor1* depletion (**Extended Data Fig. 4j,k**).

Extended Data Figure 4

Extended Data Fig. 4 | mTORC1 and LAMTOR-Rag regulate the IFN program via distinct mechanisms.

- a**, Immunoblot analysis of resting-state mTORC1 signaling in parental, *Lamtor1* KO, *Rraga* KO, and *Rragc* KO cells. Cells were lysed and analyzed for the levels of the indicated proteins and phosphorylation status of S6K1 (T389), 4E-BP1 (S65), and S6 (S235/236).
- b**, *Ifnb1* mRNA expression in RAW cells after stimulation with PRR agonists with or without rapamycin (100 nM) or Torin 1 (250 nM). The drugs were added 30 min before PRR stimulation and remained throughout. Data are mean \pm SD, $n = 3$ biological replicates per group; two-way ANOVA followed by Tukey's test, adjusted P value as indicated.
- c**, IFN- β secretion from RAW cells as treated in **b**. Data are mean \pm SD, $n = 3$ biological replicates per group; two-way ANOVA followed by Tukey's test, adjusted P value as indicated.
- d**, Immunoblot analysis of mTORC1 signaling in cells treated with mTORC1 inhibitors. Cells were lysed and analyzed for the levels of the indicated proteins and phosphorylation status of TFEB (S122), S6K1 (T389), 4E-BP1 (S65), and S6 (S235/236).
- e**, Immunoblot analysis of S6K1 phosphorylation in parental and *Rptor* KO cells following PRR stimulation.
- f**, *Ifnb1* mRNA expression in parental, *Lamtor1* KO, and *Rptor* KO cells after PRR stimulation. Data are mean \pm SD, $n = 3$ biological replicates per group; two-way ANOVA followed by Tukey's test, adjusted P value as indicated.
- g**, IFN- β secretion from parental, *Lamtor1* KO, and *Rptor* KO cells after PRR stimulation. Data are mean \pm SD, $n = 3$ biological replicates per group; two-way ANOVA followed by Tukey's test, adjusted P value as indicated.
- h**, Immunoblot analysis of the indicated ISGs in parental and *Rptor* KO cells in response to PRR stimulation.
- i**, Immunoblot analysis of S6K1 phosphorylation in parental, *Lamtor1* KO, and *Lamtor1* KO cells stably expressing Raptor-Rheb15.
- j**, *Ifnb1* mRNA expression in parental, *Lamtor1* KO, and *Lamtor1* KO cells expressing Raptor-Rheb15 after PRR stimulation. Data are mean \pm SD, $n = 3$ biological replicates per group; two-way ANOVA followed by Tukey's test, adjusted P value as indicated.
- k**, IFN- β secretion from parental, *Lamtor1* KO, and *Lamtor1* KO cells expressing Raptor-Rheb15 after PRR stimulation, as measured by ELISA. Data are mean \pm SD, $n = 3$ biological replicates per group; two-way ANOVA followed by Tukey's test, adjusted P value as indicated.

Additionally, despite mTORC1 hyperactivity (Reviewer Figure 2a-c), the significant IFN response observed in the absence of DEPDC5 could be abolished by *Mapk14/11* depletion, suggesting that mTORC1 activity alone cannot explain the phenotype.

Reviewer Figure 2

a

b

c

Reviewer Figure 2 | Signaling hierarchy of mTORC1 and p38 MAPK in regulating IFN induction.

a, Immunoblot analysis of S6K1, TFEB, and p38 MAPK phosphorylation in parental, *Mapk14/11* DKO, *Depdc5* KO, and *Depdc5/Mapk14/11* TKO cells.

b,c, *lfnb1* transcription (b) and IFN- β release (c) from parental, *Mapk14/11* DKO, *Depdc5* KO, and *Depdc5/Mapk14/11* TKO cells after PRR stimulation. Data are mean \pm SD, $n = 3$ biological replicates per group; two-way ANOVA followed by Tukey's test, adjusted P value as indicated.

- **TFEB/TFE3 as positive regulators:**

While TFEB/TFE3 are non-canonical mTORC1 substrates under the control of LAMTOR-Rag axis, their depletion (individually or together) abolished IFN induction (**Extended Data Fig. 5; Reviewer Fig. 8**), indicating they act as essential transcriptional regulators rather than suppressors downstream of LAMTOR-Rag.

Extended Data Figure 5

Extended Data Fig. 5 | Loss of TFEB/TFE3 impairs the PRR-induced IFN response.

a-c, Immunoblot analysis of TFEB phosphorylation in parental, *Lamtor1* KO, *Rraga* KO, and *Rragc* KO cells following PRR stimulation.

d, Immunoblot analysis of TFEB and TFE3 in parental, *Lamtor1* KO, *Rraga* KO, and *Rragc* KO cells using the same batch of samples as in Fig. 3g, with the same actin blot as the loading control.

e, Validation of TFE3 expression in *Tfe3* KO cells.

f,g, *ifnb1* transcription (**f**) and IFN- β release (**g**) from *Tfe3* KO cells after PRR stimulation. Data are mean \pm SD, $n = 3$ biological replicates per group; two-way ANOVA followed by Tukey's test, adjusted P value as indicated.

h, Validation of TFEB expression in parental, *Lamtor1* KO, *Tfeb* KO, and *Lamtor1/Tfeb* DKO cells.

i,j, *ifnb1* transcription (**i**) and IFN- β release (**j**) from the indicated cells after PRR stimulation. Data are mean \pm SD, $n = 3$ biological replicates per group; two-way ANOVA followed by Tukey's test, adjusted P value as indicated.

Reviewer Figure 8

a

b

Reviewer Figure 8 | TFEB/TFE3 positively regulates IFN induction.

a, Validation of TFEB/TFE3 expression in parental and *Flcn* KO, *Flcn/Tfeb* DKO, and *Tfeb/Tfe3* DKO cells.

b, *ifnb1* mRNA levels in parental, *Flcn* KO, *Flcn/Tfeb* DKO, and *Tfeb/Tfe3* DKO cells following PRR stimulation. Data are mean \pm SD, $n = 3$ biological replicates per group; two-way ANOVA followed by Tukey's test, adjusted P value as indicated.

Overall, our data suggest that mTORC1 as a primarily anabolic regulator exerts its effects on IFN production largely through protein translation. While global metabolic changes may contribute, our data highlight a *direct* signaling role for LAMTOR-Rag-p38 axis in regulating IFN production.

3. Post-transcriptional role of p38 MAPK in IFN regulation

Since the distinct Rag GTPase nucleotide states specifically affected p38 MAPK phosphorylation but not its downstream nuclear targets ATF-2 and c-Jun (**Figure 4j in the revised manuscript**), we examined p38 MAPK subcellular localization following PRR stimulation. Nuclear-cytoplasmic fractionation assays revealed predominant cytoplasmic localization of p38 MAPK, regardless of PRR stimulation or *Lamtor1* status (**new Figure 6a**). Using polymer-based magnetic bead isolation, we demonstrated the presence of p38 MAPK in phagolysosomal fractions (**Figure 6b,c**).

Building on our findings that FLCN and DEPDC5 oppositely regulate p38 phosphorylation and IFN responses (**Figure 5a-c in the revised manuscript**), we tested whether Rag GTPase nucleotide states control p38 lysosomal recruitment. *Depdc5* knockout markedly increased phagolysosomal localization of both Raptor and p38, while *Flcn* knockout abolished p38 phagolysosomal association (**Figure 6d**). Co-immunoprecipitation demonstrated a specific interaction between p38 MAPK and FLCN, requiring both Longin and DENN domains (**Figure 6e**). Supporting these observations, live-cell imaging revealed abolished

co-localization of mBaoJin-MAPK14 with LysoView-positive lysosomes in *Flcn* KO versus *Depdc5* KO or control cells (**Figure 6f**). Collectively, these data identify FLCN as a key mediator of p38 MAPK lysosomal recruitment, directly coupling Rag GTPase nucleotide status to spatially regulated p38 activation.

Figure 6

Fig. 6 | p38 MAPK is required to regulate *Ifnb1* mRNA stability.

a, Nuclear-cytoplasmic fractionation assays to determine the p38 MAPK localization in response to PRR stimulation.

b, Schematic workflow for the isolation of native phagolysosomes.

c, Immunoblot analysis of phagolysosomes isolated from parental and *Lamtor1* KO cells. WCL, whole cell lysate.

d, Immunoblot analysis of phagolysosomes isolated from parental, *Flcn* KO, and *Depdc5* KO cells. The asterisk (*) indicates a non-specific band detected by the FLCN antibody.

e, Interaction between mBaoJin-MAPK14 and FLAG-tagged mouse FLCN or the indicated variants, as assessed by FLAG immunoprecipitation in HEK293T cells.

- f, Representative micrographs showing co-localization of mBaoJin-MAPK14 with LysoView-positive lysosomes. Scale bars: 10 μ m.
- g, Measurements of *Ifnb1* mRNA decay using Roadblock-qPCR in parental, *Lamtor1* KO, and *Mapk14/11* DKO cells.
- h, Measurements of *Ifnb1* mRNA decay using Roadblock-qPCR in parental, *Depdc5* KO, *Mapk14/11* DKO, and *Depdc5/Mapk14/Mapk11* TKO cells.
- i, Representative micrographs of RAW macrophages of the indicated genotypes following infection with IAV PR8-NS1-GFP at an MOI of 0.25 for 24 h. Scale bars: 100 μ m.
- j, Schematic representation of the dual mechanism by which the lysosomal LAMTOR-Rag complex regulates IFN production.

Considering the inherent instability of *Ifnb1* mRNA, we hypothesized that extranuclear p38 MAPK activity during PRR stimulation might regulate its stability. We employed Roadblock-qPCR to measure the half-life of *Ifnb1* mRNA. Remarkably, *Ifnb1* mRNA degradation was significantly accelerated in both *Lamtor1* KO and *Mapk14/Mapk11* DKO cells relative to parental cells (**Figure 6g**). In contrast, DEPDC5 deficiency substantially prolonged *Ifnb1* mRNA half-life, an effect that was reversed by *Mapk14/Mapk11* deletion (**Figure 6h**). These results demonstrate that p38 MAPK critically regulates *Ifnb1* mRNA stability downstream of the LAMTOR-Rag GTPases.

Consistent with this post-transcriptional role of p38, impaired IFN production in *Lamtor1* KO, *Flcn* KO, and *Mapk14/11* DKO cells increased viral susceptibility, whereas the potent antiviral state in *Depdc5* KO cells was strictly dependent on p38 MAPK activity (**Figure 6i**).

In summary, we now have a clearer understanding of how lysosomal signaling is involved in regulating IFN induction during antiviral responses. The LAMTOR-Rag GTPase complex governs IFN- β production through dual transcriptional and post-transcriptional control. Mechanistically, Rag GTPase nucleotide cycling controls the IRF expression, priming the IFN transcriptional program. FLCN recruits p38 MAPK to lysosomes, where Rag nucleotide cycling dynamically activates lysosomal p38 MAPK, which is required to stabilize *Ifnb1* mRNA. Finally, nutrient cues drive Rag nucleotide cycling, thus coupling IFN production to cellular anabolic capacity.

4. Integration with prior literature

We appreciate the reviewer's emphasis on linking our findings to the established roles of LAMTOR-Rag in immunity. In response, the revised manuscript now cites and discusses the connection between LAMTOR/Ragulator and pro-inflammatory cytokines (e.g., TNF- α /IL-6), potentially mediated by TFEB/TFE3 regulation [PMID: 27731330, 27171064, 29686050].

A point to emphasize here is that mTORC1 is not directly involved in activating IFN transcription (except for TLR3 signaling) but is essential for sustaining the IFN response through global translation and protein secretion.

Together with emerging evidence [PMID: 25238095, 32433612, 39847635], our findings highlight the potential synergy between metabolic rewiring (e.g., amino acid sensing) and immune signaling, positioning lysosomes as hubs for integrating these pathways.

Key revisions addressing your concerns:

1. Removed the "LYSFII" terminology and reframed the pathway as a lysosomal signaling module.
2. Added cytokine profiling data demonstrating selectivity for IFN/ISGs (**Extended Data Fig. 2g**).

3. Clarified mTORC1/TFEB relationships with new genetic evidence (**Extended Data Fig. 5; Reviewer Figs. 2, 8**).
4. Strengthened the epistatic relationship between p38 MAPK and LAMTOR-Rag (**Fig. 5**).
5. Presented evidence showing the post-transcriptional role of p38 MAPK and its lysosomal recruitment by FLCN (**Fig. 6 and Extended Data Fig. 10**).
6. Expanded the Discussion to acknowledge the metabolic requirements and contextualize findings within lysosomal-immune crosstalk.

Thank you again for your valuable insights, which enhanced the rigor and balance of our conclusions while strengthening the overall quality and clarity of our work.

Dear Chun-Yan,

Thank you for transferring the revised version of your manuscript to The EMBO Journal. The study has now been seen by all original referees, who appreciate the revisions, but also find that several of their initial points were not sufficiently addressed or clarified.

Therefore, I would like to invite you to address the remaining referee comments either by textual changes or additional experimentation. From the editorial point of view, concerns by reviewer #2, point 2 and reviewer #2, point 1 (use of constitutively active p38) and point 3 would be important to address experimentally. Adding evidence that p38 can regulate IRF5/7 (reviewer #3, in their point 2) would also strengthen the conclusions.

I would also be happy to discuss the revisions further via email or phone/videoconference.

We generally allow three months as standard revision time. Should you foresee a problem in meeting this deadline, please let us know in advance to discuss an extension.

As a matter of policy, competing manuscripts published during this period will not negatively impact on our assessment of the conceptual advance presented by your study. However, please contact me as soon as possible upon publication of any related work to discuss the appropriate course of action.

When preparing your letter of response to the referees' comments, please bear in mind that this will form part of the Review Process File and will therefore be available online to the community. For more details on our Transparent Editorial Process, please visit our website: <https://www.embopress.org/page/journal/14602075/authorguide#transparentprocess>. Please also see the attached document with instructions for guidelines on preparation of the revised manuscript.

Please feel free to contact me if you have any questions regarding this final revision. Thank you again for giving us the chance to consider your manuscript for The EMBO Journal. I look forward to receiving the revised version.

With best wishes,

Ieva

We realize that it is difficult to revise to a specific deadline. In the interest of protecting the conceptual advance provided by the work, we recommend a revision within 3 months (28th Jan 2026). Please discuss the revision progress ahead of this time with the editor if you require more time to complete the revisions.

Referee #1:

I appreciate the authors' extensive revision of the manuscript. They have satisfactorily addressed my comments. However, one outstanding issue is clearly stating the caveats in the discussion to clarify that this pathway has other effects.

I don't have to re-review this manuscript, and I am supportive of this work. It should be published.

Referee #2:

The revised version of the manuscript demonstrates substantial improvement. The new data and textual adjustments have increased its clarity and impact, and the majority of my earlier comments have been thoroughly addressed. Kudos to the authors for producing such a strong piece of work. Only a few outstanding issues remain to be resolved prior to acceptance.

1) Extended Data Figures 4a and 4i: Instead of (or in addition to) the phosphorylation of canonical mTORC1 substrates (like S6K and 4EBP1) the authors should also include the phosphorylation of the lysosomal substrate TFEB to assess the relative effects of each genetic or pharmacological perturbation to lysosomal and non-lysosomal substrates. In fact, phosphorylation of TFEB is more relevant for the characterization of Rag- or LAMTOR-related mutant cell lines (as also described recently, eg PMIDs: 32612235 and 39385049).

2) Reviewer Figure 3 (about the effects of osmostress and arsenite) is an important control experiment and should be included in the revised manuscript as an ED Figure with appropriate references in the text. Please also specify the timing of LPS treatment in these experiments. Is this comparable to that of osmostress and arsenite (4h)?

3) Reviewer Figure 5 (lysosomal localization of mBaoJin-MAPK14 upon starvation) is a nice control experiment and should be included as an ED Figure with appropriate references in the text.

Referee #3:

The authors addressed most of my previous concerns and significantly improved their manuscript. However, some key issues remain unresolved:

1. The involvement of p38 in the control of IFN-beta remains unresolved. PRR stimulation appears to inhibit, rather than promote, p38 phosphorylation (Extended Data Fig 9e-g), which argues against a role for p38 activation in promoting antiviral responses. Furthermore, although the authors initially postulate that p38 activation mediates IFN-beta production via AP-1, this hypothesis was not experimentally tested. Instead, they suggest that p38 controls *Inf1* mRNA stability at the post-transcriptional level. However, the rationale for this hypothesis and the mechanism of this putative p38 function remain unclear. Even more puzzlingly, the authors report that p38 is recruited to lysosomes by FLCN, but the function of p38 upon lysosomal recruitment remains obscure. As a result, it remains uncertain whether, and by what mechanism, p38 physiologically regulates the interferon response.

Finally, to conclusively prove that p38 acts downstream of the Rag GTPases, the authors should express a constitutively active form of p38 in LAMTOR/Rag-KO cells and assess *Inf1* expression.

2. The authors show that LAMTOR/Rag depletion strongly reduces IRF5/7 expression (Fig. 3f-i). However, they do not assess whether this observation is functionally relevant to IFN regulation. Does IRF5/7 re-expression rescue *Ifnb1* expression and IFN-beta secretion in LAMTOR/Rag-KO cells? Does starvation affect IRF5/7 levels? Does p38 regulate IRF5/7, and can IRF5/7 overexpression rescue IFN-beta expression in MAPK14- or MAPK11-deficient cells?

3. The observation that LAMTOR/Rag deficiency leads to a strong reduction in TFEB/TFE3 protein levels (Fig. 1h and Extended Dat Fig. 5a-d) is not in line with robust literature showing that LAMTOR/Rag-KO leads to TFEB/TFE3 nuclear translocation without significantly affecting their expression levels. The apparent decrease in TFEB/TFE3 protein levels may therefore result from incomplete lysis of nuclear fractions, leading to loss of nuclear material and an artifactual reduction in detectable protein. To assess this possibility, the authors should use more stringent lysis conditions (e.g. RIPA buffer followed by sonication), to ensure complete nuclear protein extraction. A similar concern also applies to the reduction of IRF5/7 levels upon LAMTOR/Rag depletion (see point 2), which should also be validated using stronger lysis conditions and by assessing IRF5/7 subcellular localization, via immunofluorescence or cell fractionation analyses.

Point-by-point response to EMBO Journal submission EMBOJ-2025-122531-T

We sincerely thank all the reviewers for their constructive comments and insightful suggestions, which have helped us further enhance the overall impact and clarity of our work. We have completed the necessary experiments and incorporated the results into the revised manuscript, and made text modifications where necessary.

Below, we provide detailed point-by-point responses to all comments.

Reviewers' comments:

Referee #1:

I appreciate the authors' extensive revision of the manuscript. They have satisfactorily addressed my comments. However, one outstanding issue is clearly stating the caveats in the discussion to clarify that this pathway has other effects.

I don't have to re-review this manuscript, and I am supportive of this work. It should be published.

We thank Reviewer #1 for approving our manuscript for publication. We have included a discussion on the pleiotropic roles of the LAMTOR-Rag signaling module in cellular homeostasis to the revised manuscript (lines 454-459).

Referee #2:

The revised version of the manuscript demonstrates substantial improvement. The new data and textual adjustments have increased its clarity and impact, and the majority of my earlier comments have been thoroughly addressed. Kudos to the authors for producing such a strong piece of work. Only a few outstanding issues remain to be resolved prior to acceptance.

1) Extended Data Figures 4a and 4i: Instead of (or in addition to) the phosphorylation of canonical mTORC1 substrates (like S6K and 4EBP1) the authors should also include the phosphorylation of the lysosomal substrate TFEB to assess the relative effects of each genetic or pharmacological perturbation to lysosomal and non-lysosomal substrates. In fact, phosphorylation of TFEB is more relevant for the characterization of Rag- or LAMTOR-related mutant cell lines (as also described recently, eg PMIDs: 32612235 and 39385049).

We have already performed the immunoblot analysis for all mTORC1 substrates during our previous revisions. We are pleased to include the phospho-TFEB and total-TFEB blots (from the same experimental batch) in the revised **Figures EV4A and EV4I**.

Figure EV4

Figure EV4. mTORC1 and LAMTOR-Rag regulate the IFN program via distinct mechanisms.

(A) Immunoblot analysis of resting-state mTORC1 signaling in parental, *Lamtor1* KO, *Rraga* KO, and *Rragc* KO cells. Cells were lysed and analyzed for the levels of the indicated proteins and phosphorylation status of TFE8 (S122), S6K1 (T389), 4E-BP1 (S65), and S6 (S235/236).

(I) Immunoblot analysis of TFE8 and S6K1 phosphorylation status in parental, *Lamtor1* KO, and *Lamtor1* KO cells stably expressing Raptor-Rheb15.

2) Reviewer Figure 3 (about the effects of osmotic stress and arsenite) is an important control experiment and should be included in the revised manuscript as an ED Figure with appropriate references in the text. Please also specify the timing of LPS treatment in these experiments. Is this comparable to that of osmotic stress and arsenite (4h)?

In these experiments, cells were treated with sorbitol (0.5 M), sodium arsenite (250 μM), and LPS (0.5 $\mu\text{g ml}^{-1}$) for 4 hours before harvest. We have now included the data in the revised Figures EV10E and EV10F with references in the text (lines 361-363).

Figure EV10

Figure EV10. p38 MAPK activation is critical for PRR-induced IFN induction.

(E) Immunoblot analysis of p38 MAPK phosphorylation in parental and *Mapk14/11* DKO cells following 4 h of stimulation with sorbitol (0.5 M) or sodium arsenite (250 μM).

(F) *Ifnb1* transcription (left) and IFN- β release (right) from parental, *Mapk14/11* DKO, and *Lamtor1* KO cells after 4 h of treatment with sorbitol, sodium arsenite, and LPS. Data are mean \pm SD, $n = 3$ biological replicates per group; two-way ANOVA followed by Tukey's test, adjusted P value as indicated.

3) Reviewer Figure 5 (lysosomal localization of mBaoJin-MAPK14 upon starvation) is a nice control experiment and should be included as an ED Figure with appropriate references in the text.

We have included the data in the revised Figure EV10K with references in the text (lines 402-404).

Figure EV10

Figure EV10. p38 MAPK activation is critical for PRR-induced IFN induction.

(K) Representative micrographs showing abolished co-localization of mBaoJin-MAPK14 with LysoView-positive lysosomes in response to amino acid starvation. Scale bars: 10 μm .

The authors addressed most of my previous concerns and significantly improved their manuscript. However, some key issues remain unresolved:

1. The involvement of p38 in the control of IFN-beta remains unresolved. PRR stimulation appears to inhibit, rather than promote, p38 phosphorylation (Extended Data Fig 9e-g), which argues against a role for p38 activation in promoting antiviral responses. Furthermore, although the authors initially postulate that p38 activation mediates IFN-beta production via AP-1, this hypothesis was not experimentally tested. Instead, they suggest that p38 controls *Inf1* mRNA stability at the post-transcriptional level. However, the rationale for this hypothesis and the mechanism of this putative p38 function remain unclear. Even more puzzlingly, the authors report that p38 is recruited to lysosomes by FLCN, but the function of p38 upon lysosomal recruitment remains obscure. As a result, it remains uncertain whether, and by what mechanism, p38 physiologically regulates the interferon response.

Finally, to conclusively prove that p38 acts downstream of the Rag GTPases, the authors should express a constitutively active form of p38 in LAMTOR/Rag-KO cells and assess *Inf1* expression.

The induction of IFN in response to viral infection is canonically understood to be initiated by PRR signaling. This pathway involves the activation of TBK1/IKK kinases, which phosphorylate IRFs, prompting their nuclear translocation and the subsequent transcription of IFN genes.

Our study uncovers a parallel and essential regulatory axis for the IFN response, orchestrated by the lysosomal LAMTOR-Rag signaling complex. This pathway operates independently of direct PRR signal transduction.

Instead, the LAMTOR-Rag complex, through its distinct nucleotide-binding states, serves to pre-condition the IFN transcriptional program by modulating the expression levels of key IRFs. Furthermore, we identify a mechanism for post-transcriptional regulation: the tumor suppressor FLCN recruits p38 MAPK to the lysosomal surface, where Rag GTPase nucleotide cycling dynamically activates this pool of p38. This lysosome-localized p38 activation is functionally required for stabilizing *Inf1* mRNA, ensuring robust IFN- β production.

Critically, the parallel importance of both transcriptional and post-transcriptional regulation is demonstrated by the finding that overexpression of a constitutively active p38 mutant in LAMTOR-Rag deficient cells yielded only a slight rescue of IFN responses (**Figure EV11D-F; lines 422-425**). This confirms that both the LAMTOR-Rag-mediated priming of IRF expression and the Rag-p38-mediated mRNA stabilization are non-redundant and necessary for a full IFN response.

Notably, the nutrient-responsive Rag nucleotide cycle integrates cellular anabolic capacity with IFN production, suggesting a metabolic checkpoint for this immune response. We propose a model wherein Rag nucleotide states recruit upstream regulators to spatiotemporally control p38 phosphorylation at the lysosome, although the precise mechanisms remain a subject for future investigation. Importantly, we rule out a role for the canonical p38-AP-1 axis in this process, as LAMTOR-Rag ablation did not affect AP-1 phosphorylation (ATF-2/c-Jun), and p38 MAPK did not translocate to the nucleus upon PRR activation, irrespective of LAMTOR status.

Figure EV11

Figure EV11. Parallel importance of transcriptional activation and post-transcription regulation for IFN production.

(D) Ectopic expression of the constitutively active p38 (D176A/F327S) in LAMTOR-Rag deficient cells.

(E,F) *lfnb1* transcription (E) and IFN- β release (F) from parental, LAMTOR-Rag deficient cells or LAMTOR-Rag deficient cells stably expressing the constitutively active p38 after PRR stimulation. Data are mean \pm SD, $n = 3$ biological replicates per group; two-way ANOVA followed by Tukey's test, adjusted P value as indicated.

2. The authors show that LAMTOR/Rag depletion strongly reduces IRF5/7 expression (Fig. 3f-i). However, they do not assess whether this observation is functionally relevant to IFN regulation. Does IRF5/7 re-expression rescue *lfnb1* expression and IFN-beta secretion in LAMTOR/Rag-KO cells? Does starvation affect IRF5/7 levels? Does p38 regulate IRF5/7, and can IRF5/7 overexpression rescue IFN-beta expression in MAPK14- or MAPK11-deficient cells?

Overexpression of IRF5/7 in *Lamtor1* KO cells only partially restored *lfnb1* expression and IFN- β secretion (Figure 4C-E; lines 334-335). Combined with the results from active p38 overexpression experiments (Figure EV11D-F; lines 422-425; see response to Q1), these data indicate that the magnitude of IFN output is determined by both the transcriptional activation of *lfnb1* (governed by IRF abundance) and the post-transcriptional stability of its mRNA (mediated by p38 MAPK). The LAMTOR-Rag signaling pathway appears to regulate both of these arms.

Figure 4

Figure 4. Loss of LAMTOR-Rag function abolishes p38 MAPK activation.

(C) Ectopic expression of IRF5 and IRF7 in *Lamtor1* KO cells.

(D,E) Induction of *Ifnb1* transcript (D) and IFN- β release (E) from parental, *Lamtor1* KO, and *Lamtor1* KO cells stably expressing IRF-5 and IRF-7 after PRR stimulation. Data are mean \pm SD, $n = 3$ biological replicates per group; two-way ANOVA followed by Tukey's test, adjusted P value as indicated.

A comprehensive mechanistic dissection of IRF5/7 regulation by the lysosomal pathway is the primary focus of a separate, ongoing study in our lab. To protect the novelty and scope of that work, we have chosen not to include those extensive mechanistic insights here. Our present data suggest that MAPK14/11 is not critically involved, implying that LAMTOR-Rag signaling may regulate IRF5/7 through an independent mechanism.

3. The observation that LAMTOR/Rag deficiency leads to a strong reduction in TFEB/TFE3 protein levels (Fig. 1h and Extended Dat Fig. 5a-d) is not in line with robust literature showing that LAMTOR/Rag-KO leads to TFEB/TFE3 nuclear translocation without significantly affecting their expression levels. The apparent decrease in TFEB/TFE3 protein levels may therefore result from incomplete lysis of nuclear fractions, leading to loss of nuclear material and an artifactual reduction in detectable protein. To assess this possibility, the authors should use more stringent lysis conditions (e.g. RIPA buffer followed by sonication), to ensure complete nuclear protein extraction. A similar concern also applies to the reduction of IRF5/7 levels upon LAMTOR/Rag depletion (see point 2), which should also be validated using stronger lysis conditions and by assessing IRF5/7 subcellular localization, via immunofluorescence or cell fractionation analyses.

We thank Reviewer #3 for their helpful suggestion to evaluate TFEB/TFE3 protein expression under more stringent lysis conditions. As shown in **Reviewer Figure 1**, RIPA buffer combined with sonication [B] enhanced the extraction of nuclear TFEB/TFE3 compared to our standard S6K lysis buffer (containing 1% Triton X-100) [A]. Importantly, this stringent lysis did not alter the levels of phosphorylated TFEB (which is predominantly cytosolic), confirming the specificity of the extraction.

This phenomenon is consistent with previous findings using similar non-ionic detergent (1% Nonidet P-40), as reported by the laboratory of Masato Okada (PMID: 35183507), and we have now addressed this lysis-dependent discrepancy in the revised manuscript (**lines 241-243**). Critically, the significant reduction in IRF5/7 expression was consistently observed under both lysis conditions, thereby ruling out incomplete extraction as a confounding factor for this key finding.

Reviewer Figure 1

Reviewer Figure 1. Evaluation of protein extraction stringency using S6K lysis buffer compared to RIPA lysis buffer followed by sonication.

(A,B) Protein expression of TFEB/TFE3 and IRF5/7 in LAMTOR-Rag deficient cells relative to parental cells under S6K lysis buffer (A) and under RIPA lysis buffer conditions plus sonication (B).

Dear Chun-Yan,

Thank you for submitting a revised version of your manuscript. I apologise for the delay in the handling of your revision due to the high number of submissions and revisions that we receive at the moment. I have now gone through your responses to the remaining reviewers' comments, and I find them generally reasonable.

There now remain only a few editorial and formatting aspects as outlined below that still need to be implemented in the manuscript before its acceptance:

1. Please check that the funding information is correct and identical both in the manuscript and our online system. Currently, the National High-level Talents Program of China, the Overseas Experts Supporting Programs in China, and the R&D Program of Guangzhou National Laboratory (Grants SRPG22-002 and GZNL2023A02004) are mentioned in the Acknowledgments but are not included in the online submission system.
2. CRedit has replaced the traditional author contributions section because it offers a systematic, machine-readable author contributions format that allows for more effective research assessment. Please remove the Authors Contributions from the manuscript and use the free text boxes beneath each contributing author's name in our online submission system to add specific details on the author's contribution. More information is available in our guide to authors.
3. Please remove EV Table and Dataset legends from the manuscript text file and add to the table/dataset files in separate tab/worksheet.
4. Thank you for providing the information on the sensitivity limits of the used microplate reader, leading to some repetitions in the numerical values. While keeping this in mind, our standard numerical data analysis suggests that the mock samples may be reused between several experimental conditions in a number of figure panels (please see the attached files with the detected duplications/repetitions in colour). If this is the case, this should be clearly indicated in the appropriate figure legends.
5. In our standard image integrity check, we noted that the actin panel in Figures EV8A and B appear to be rather similar. Please check and provide the source data for these figures.
6. Our data editors have flagged the following issues in figure legends that need correcting:
 - Please provide the exact p values in the legends of figures 1G, I, M; 2A, B; 3A-D; 4D, E; 5B, C, E, F; EV1 A, D, E; EV3 F, H; EV4 B, C, F, G, J, K; EV5 F, H, A, I; EV6 B, C; EV7 B, C, E, F, G, H, I, J, K; EV8 B, D, E, G, H, I, J, K; EV8 L, N; EV10 A, B, F, H, I; EV11 B, C, D.
 - Please indicate the statistical test used for data analysis in the legends of figures 4G, H; EV8 I.
 - Please provide information on the number and nature of replicates in the legends of figures 1B, 6G, H, EV8 I.
 - Please define the error bars in the legends of figures 1B, 2H, I, J; 6G, H.
7. Papers published in The EMBO Journal are accompanied online by a 'Synopsis' to enhance discoverability of the manuscript. It consists of A) a short (1-2 sentences) summary of the findings and their significance, B) 3-4 bullet points highlighting key results and C) a synopsis image that is 550x300-600 pixels large (width x height, jpeg or png format). You can either show a model or key data in the synopsis image. Please note that the image size is rather small and that text needs to be readable at the final size. Please send us this information together with the revised manuscript.

Please feel free to contact me if have any questions regarding these final points. I look forward to your receiving the updated manuscript.

With best wishes,

Ieva

We realize that it is difficult to revise to a specific deadline. In the interest of protecting the conceptual advance provided by the work, we recommend a revision within 3 months (18th Mar 2026). Please discuss the revision progress ahead of this time with the editor if you require more time to complete the revisions. Use the link below to submit your revision:

The authors addresses the remaining editorial issues.

Dear Chun-Yan,

Thank you for addressing the final editorial points. I am now pleased to inform you that your manuscript has been accepted for publication in the EMBO Journal.

Before we forward your manuscript to our publishers, we would like to propose some edits in the manuscript title, abstract and synopsis (please see the attached file). I have also prepared a short blurb that will accompany the title of your manuscript in our online system. Please take a look and let me know if any corrections are needed.

Please note that it is The EMBO Journal policy for the transcript of the editorial process (containing referee reports and your response letters) to be published as an online supplement to each paper. If you should prefer removal of any referee-only figures included in the point-by-point response(s), e.g. because they may still be used for future publication or because they have been reproduced from published work by others, please do let us know immediately via response email.

More information is available here: <https://link.springer.com/partners/embo-press/editorial-policies#Peer%20review>

You may qualify for financial assistance for your publication charges - either via a Springer Nature fully open access agreement or an EMBO initiative. Check your eligibility: <https://link.springer.com/journal/44318/how-to-publish-with-us>

If you have any questions, please do not hesitate to contact the Editorial Office or me directly. Thank you for this contribution to The EMBO Journal and congratulations on a nice study!

With best wishes,

Ieva
